# Fine-Tuning Personalization in Federated Learning to Mitigate Adversarial Clients

**Youssef Allouah**[*]
EPFL
Lausanne, Switzerland

**Abdellah El Mrini**[†]
EPFL
Lausanne, Switzerland

**Rachid Guerraoui**
EPFL
Lausanne, Switzerland

**Nirupam Gupta**
University of Copenhagen
Copenhagen, Denmark

**Rafael Pinot**
Sorbonne Université and Université Paris Cité,
CNRS, F-75005 Paris, France

## Abstract

Federated learning (FL) is an appealing paradigm that allows a group of machines (a.k.a. clients) to learn collectively while keeping their data local. However, due to the heterogeneity between the clients' data distributions, the model obtained through the use of FL algorithms may perform poorly on some client's data. Personalization addresses this issue by enabling each client to have a different model tailored to their own data while simultaneously benefiting from the other clients' data. We consider an FL setting where some clients can be adversarial, and we derive conditions under which full collaboration fails. Specifically, we analyze the generalization performance of an interpolated personalized FL framework in the presence of adversarial clients, and we precisely characterize situations when full collaboration performs strictly worse than *fine-tuned* personalization. Our analysis determines how much we should scale down the level of collaboration, according to data heterogeneity and the tolerable fraction of adversarial clients. We support our findings with empirical results on mean estimation and binary classification problems, considering synthetic and benchmark image classification datasets.

## 1  Introduction

Federated learning (FL) is the de facto standard for a group of machines (also referred to as *clients*) to learn a common model on their collective data (Kairouz et al., 2021). The benefit of this method is twofold: the clients (1) retain control over their local data (as they do not communicate them to a central server) and (2) share the computational workload during the learning procedure. Although FL is a promising learning paradigm, it also comes with its own technical challenges, the first of which is data heterogeneity. Indeed, each client holds only a portion of the common dataset, which is not necessarily a faithful representation of the entire population. This could lead to disagreements amongst clients on model updates, rendering the task of learning an accurate model in this context cumbersome. Nevertheless, when all machines correctly follow a prescribed algorithm, this problem can be solved using either a distributed implementation of the stochastic gradient descent (SGD) algorithm or variants of the federated averaging scheme (McMahan et al., 2017; Khaled et al., 2020; Karimireddy et al., 2020). Another negative impact of data heterogeneity is the non-uniformity of model accuracy. Specifically, while an FL algorithm outputs a model that is accurate on average, the accuracy of this model on different clients' local data could vary significantly.

---

[*]Authors are listed in alphabetical order.
[†]Correspondence to : Abdellah El Mrini <abdellah.elmrini@epfl.ch>.

38th Conference on Neural Information Processing Systems (NeurIPS 2024).

**Personalization.** Personalized FL attempts to address the aforementioned shortcoming on non-uniform model accuracy by having clients collaborate to design individual models that generalize well over their local data distributions (Vanhaesebrouck et al., 2017; Sattler et al., 2020; Fallah et al., 2020; Hanzely et al., 2020). This approach is highly relevant to many modern machine learning applications, as it tailors the training of each model to the needs of the respective client. To get a more precise view of this approach, consider a general supervised learning task with $\mathcal{X}$ denoting the input space and $\mathcal{Y}$ the output space. Personalized FL involves $n$ clients which can communicate, through a central trusted machine (a.k.a. the *server*). The clients hold local datasets $S_1, \ldots, S_n$, comprising $m$ data points, drawn i.i.d. from their respective local data distributions $\mathcal{D}_1, \ldots, \mathcal{D}_n$ with support in $\mathcal{Z} := \mathcal{X} \times \mathcal{Y}$. Given a parameter set $\Theta \subseteq \mathbb{R}^d$, we consider hypothesis functions of the form $h_\theta \colon \mathcal{X} \to \mathcal{Y}'$, parameterized by $\theta \in \Theta$, and we denote by $\mathcal{H}$ the corresponding hypothesis class. We allow $\mathcal{Y}' \neq \mathcal{Y}$ to encompass cases in which the model outputs logits/probits instead of classes. The objective of each client $i$ is to find a model $\theta_i^* \in \Theta$ minimizing its local risk function

$$\mathcal{R}_i(\theta) := \mathbb{E}_{(x,y) \sim \mathcal{D}_i} \left[ \ell(h_\theta(x), y) \right], \forall \theta \in \Theta, \tag{1}$$

where $\ell \colon \mathcal{Y}' \times \mathcal{Y} \to \mathbb{R}_+$ is a non-negative point-wise loss function measuring how well a hypothesis $h_\theta$ fits the output $y$ on an input $x$. The mapping $(\theta, x, y) \mapsto \ell(h_\theta(x), y)$ is assumed to be differentiable, with respect to $\theta$. To minimize the local risk above, each client $i$ only has access to its local dataset $S_i$ yielding a local empirical loss $\mathcal{L}_i(\theta) := \frac{1}{m} \sum_{(x,y) \in S_i} \ell(h_\theta(x), y)$, and to the information sent by other clients on their respective datasets. This optimization problem can be solved in several ways, and we have at out disposal a rich literature on personalized FL schemes (Mansour et al., 2020; Fallah et al., 2020; Marfoq et al., 2021). We focus on a specific personalization scheme wherein each client aims to solve for the *$\lambda$-interpolated empirical loss* minimization problem:

$$\min_{\theta_i \in \Theta} \mathcal{L}_i^\lambda(\theta_i) := (1 - \lambda)\mathcal{L}_i(\theta_i) + \lambda\mathcal{L}(\theta_i), \tag{2}$$

where $\mathcal{L}(\theta) := \frac{1}{n} \sum_{i=1}^n \mathcal{L}_i(\theta)$, $\forall \theta \in \Theta$, and $\lambda \in [0, 1]$. When $\lambda = 1$, the problem reduces to the standard FL objective and corresponds to a full collaboration amongst the clients. The case of $\lambda = 0$ represents the absence of collaboration, i.e., each client minimizes its local empirical loss. We use the terminology *fine-tuned* personalization to refer to the objective of solving for (2) when $\lambda \in (0, 1)$ is determined by optimizing the learning performance in retrospect.

**Robustness.** Standard personalized FL approaches usually assume that all clients correctly follow a prescribed protocol, and hence do not consider the possibility of *adversarial* clients. Robustness to such clients, a.k.a "Byzantine robustness", refers to the design of FL schemes that yield accurate models even when some clients arbitrarily deviate from the algorithm and can send potentially corrupted information to other clients: This could result from poisonous data or code, or attacks from malicious players. Although Byzantine robustness has received significant attention in the FL community (Guerraoui et al., 2024), it remains largely understudied in the context of personalized FL. In fact, prior work has shown that the presence of adversarial clients induces a fundamental optimization error that grows with the heterogeneity across clients' local data (Karimireddy et al., 2022; Allouah et al., 2024). This optimization error is at odds with the improvement in generalization, which is offered by collaborating with other correct clients, to a point where collaboration is rendered vacuous. The main motivation for our work is to quantify the impact of adversarial clients on the relevance of collaboration for any correct client.

## 1.1 Problem Setting

Given the set of $n$ clients, we aim to tolerate the presence of at most $f < n/2$ adversarial clients. Such clients can respond arbitrarily to the queries made by the server, e.g., a gradient computation on their local dataset. We also call these clients Byzantine adversaries. The identity of Byzantine adversaries is a priori unknown to the correct (i.e., non-adversarial) clients, otherwise, the problem is rendered trivial. As we explain above, when no client is adversarial, the objective of each client $i$ is to minimize its own risk, defined in (1), by solving for the interpolated empirical loss minimization problem (2). However, in the presence of adversarial clients, a correct client cannot simply seek a solution to (2), as they can never have truthful access to the information about Byzantine adversaries' datasets. A more reasonable goal is to minimize an interpolation between their local and the *correct* clients' average loss functions. We formally define the *robust-interpolated* objective for client $i$ as follows:

$$\min_{\theta_i \in \Theta} \mathcal{L}_i^\lambda(\theta_i) := (1 - \lambda)\mathcal{L}_i(\theta_i) + \lambda\mathcal{L}_{\mathcal{C}}(\theta_i), \tag{3}$$

where $\mathcal{C}$ represents the set of correct clients, $\mathcal{L}_\mathcal{C}(\theta) := \frac{1}{|\mathcal{C}|} \sum_{i \in \mathcal{C}} \mathcal{L}_i(\theta)$, $\forall \theta \in \Theta$, and $\lambda \in [0, 1]$ is called the *collaboration level*. Similar to (2), $\lambda = 0$ and $\lambda = 1$, respectively, reduce to local learning which can be solved with local SGD (Stich, 2018), and Byzantine-robust FL which can be tackled using a robust variant of distributed gradient descent (Allouah et al., 2023). Recall that $\mathcal{C}$ is a priori unknown to the correct clients, hence they can neither have access to $\mathcal{L}_\mathcal{C}$ nor to its gradients. A correct client can only approximate any information on $\mathcal{L}_\mathcal{C}$, where the approximation error grows with both $f/n$ and data heterogeneity, shown in (Karimireddy et al., 2022; Allouah et al., 2024).

## 1.2 Contributions

We establish optimization and generalization guarantees on the interpolated personalized objective (3) in the presence of adversarial clients, and show how the degree of collaboration $\lambda$, asymptotically navigates the trade-off between the fundamental optimization error due to adversarial clients and the improved generalization performance thanks to the collaboration with correct ones.

This sheds light on specific situations in which full collaboration performs strictly worse than *fine-tuned* personalization, and even sometimes worse than local learning. Precisely, we extend an important result from domain adaptation (Ben-David et al., 2010), showing that data from a distinct distribution cannot be beneficial for the local learning task when the heterogeneity is above a certain threshold, which depends on the local sample size and the hypothesis class complexity. In the context of Byzantine robust distributed learning, we show that even if data heterogeneity is low enough, the level of collaboration should be rescaled by a parameter that both depends on the fraction of adversaries and the gradient dissimilarity between correct clients. Essentially, while the effect of collaboration could be captured by substraction between the local task complexity and the heterogeneity of local distributions in a Byzantine-free context, we show that it is further captured by a multiplicative factor in the presence of adversaries. The higher the dissimilarity between correct gradients (and the number of adversarial clients in the system), the smaller $\lambda$ should be taken.

Our results show that for personalized FL, in most realistic contexts where heterogeneity between clients is not negligible, full collaboration is not optimal in the presence of adversarial clients. Moreover, the presence of these adversarial clients considerably limits the level of collaboration that could be useful for the correct clients, to the point where it is often more efficient to learn locally if the local task is sufficiently simple.

## 1.3 Paper Outline

The remainder of the paper is organized as follows. Section 2 presents the mean estimation setting as a warm-up and special case of the framework presented in Section 1.1. The goal is to give an intuition on settings in which personalization can or cannot help reduce the estimation error in the presence of adversarial clients. Section 3 presents our analysis in the general binary classification setting, where we quantify the tension between the optimization error and the generalization gap in the presence of Byzantine adversaries. We also empirically validate our theory on the MNIST dataset. We defer full proofs to Appendix C and our full experimental setup to Appendix D. We also include further information on related work in Appendix A.

## 2 Warm-up: Mean Estimation with Adversaries

Before diving into the details of our results in a general supervised learning setting, let us first focus on the simple task of Byzantine-robust federated mean estimation in which the concepts of heterogeneity and task complexity are easier to grasp. We consider $n - f$ correct clients denoted $\mathcal{C}$. Each client $i \in \mathcal{C}$ has access to $m$ data point $y_i^{(1)}, ..., y_i^{(m)}$ sampled i.i.d. from their a local distribution $\mathcal{D}_i$ and independently from the other correct clients. We assume that each local distribution $\mathcal{D}_i$ has a support in $\mathbb{R}^d$, an unknown finite mean $\mu_i$. Furthermore, we assume that all the distributions have the same finite variance $\sigma^2 := \mathbb{E}_{y \sim \mathcal{D}_i}\left[\|y - \mu_i\|^2\right]$. In this context, the objective of each client $i$ is to find their true mean $\mu_i$. To do so, we assume that each client attempts solve a problem of the form

$$\min_{y_i^\lambda \in \mathbb{R}^d} \frac{1-\lambda}{m} \sum_{k \in [m]} \left\| y_i^\lambda - y_i^{(k)} \right\|^2 + \frac{\lambda}{m(n-f)} \sum_{k \in [m]} \sum_{j \in \mathcal{C}} \left\| y_i^\lambda - y_j^{(k)} \right\|^2. \qquad (4)$$

In fact, the optimal solution to this problem is $y_i^{\lambda*} := (1-\lambda)\widehat{y}_i + \lambda\widehat{y_\mathcal{C}}$ where $\widehat{y}_i := \frac{1}{m}\sum_{j=1}^m y_i^{(j)}$ and $\widehat{y_\mathcal{C}} := \frac{1}{n-f}\sum_{i\in\mathcal{C}}\widehat{y}_i$. Recall that due to the presence of Byzantine adversaries, correct clients cannot compute $\widehat{y_\mathcal{C}}$ directly, but should instead use a robust aggregation rule $F$ to estimate $\widehat{y_\mathcal{C}}$. Specifically, we consider the natural robust estimator of $\widehat{y}_i^*$ [3], where $\widehat{y_\mathcal{C}}$ is replaced by a robust approximation

$$y_i^\lambda = (1-\lambda)\widehat{y}_i + \lambda F(\widehat{y}_1, \ldots, \widehat{y}_n). \tag{5}$$

To formalize how good a robust aggregation rule is, we use the notion of $(f, \kappa)$-robustness (Allouah et al., 2023), which we state below.

**Definition 1** $((f, \kappa)\text{-robustness})$. Let $f < n/2$ and $\kappa \geq 0$. An aggregation rule $F$ is said to be $(f, \kappa)$-*robust* if for any vectors $v_1, \ldots, v_n \in \mathbb{R}^d$, and any set $\mathcal{U} \subseteq [n]$ of size $n - f$,

$$\|F(v_1, \ldots, v_n) - \overline{v}_\mathcal{U}\|^2 \leq \frac{\kappa}{n-f}\sum_{i\in\mathcal{U}}\|v_i - \overline{v}_\mathcal{U}\|^2, \text{ with } \overline{v}_\mathcal{U} := \frac{1}{n-f}\sum_{i\in\mathcal{U}} v_i.$$

### 2.1 On the Effect of Collaboration

We evaluate the performance of the estimator $\widehat{y}_i$ by the mean squared error defined as $\mathbb{E}\left[\|y_i^\lambda - \mu_i\|^2\right]$, where the expectation is taken over the random samples $y_i^{(1)}, ..., y_i^{(m)}$ drawn i.i.d. from $\mathcal{D}_i$ and independently for every $i \in \mathcal{C}$. Then the following holds.

**Proposition 1.** *Consider the mean estimation setting described. For any $i \in \mathcal{C}$, let $y_i^\lambda$ be as defined in (5) with an aggregation rule $F$ that satisfies $(f, \kappa)$-robustness. Then the following holds true:*

$$\mathbb{E}\left[\|y_i^\lambda - \mu_i\|^2\right] \leq 3\left(1 - \frac{1}{n-f}\right)\frac{\sigma^2\Gamma(\lambda,\kappa)}{m} + 3\lambda^2(\|\mu_i - \overline{\mu}_\mathcal{C}\|^2 + \kappa\Delta^2).$$

*with $\overline{\mu}_\mathcal{C} := \frac{1}{n-f}\sum_{i\in\mathcal{C}}\mu_i$, $\Delta^2 := \frac{1}{n-f}\sum_{j\in\mathcal{C}}\|\mu_j - \overline{\mu}_\mathcal{C}\|^2$, and $\Gamma(\lambda, \kappa) := \lambda^2(\kappa+1) - 2\lambda + \frac{n-f}{n-f-1}$.*

**Tightness of the bound.** While the right-hand side of Proposition 1 may not be tight in general, we note that it is tight for $\lambda = 0$ and $\lambda = 1$, in the homogeneous setting ($\Delta = 0$), provided that we use robust averaging with $\kappa \in \mathcal{O}(f/n)$. Indeed, when $\lambda = 0$ the squared error in estimating the mean of a distribution with variance $\sigma^2$ from $m$ i.i.d. samples is in $\Omega(\sigma^2/m)$, refer (Wu, 2017). Furthermore, when $\lambda = 1$, the squared error in estimating the mean of a distribution with variance $\sigma^2$ is in $\Omega(\frac{f+1}{n}\frac{\sigma^2}{m})$, refer (Zhu et al., 2023).

**Interpretation.** Note that, in the above, the terms that represent the hardness of the local mean estimation task and the heterogeneity among correct clients are $\frac{\sigma^2}{m}$, and $\|\mu_i - \overline{\mu}_\mathcal{C}\|^2 + \kappa\Delta^2$ respectively. Indeed $\sigma^2$ is the variance of each distribution, hence the local mean estimation task is essentially as hard as $\sigma^2$ is large with respect to the number of points $m$ each client has access to. Similarly, $\Delta^2$ essentially computes how far apart the true means of the correct clients are on average, and $\|\mu_i - \overline{\mu}_\mathcal{C}\|^2$ penalizes especially the distance to the average of the correct clients for the one we observe (i.e., $i \in \mathcal{C}$). By minimizing the right-hand side of Proposition 1, we get

$$\lambda^* = \frac{\left(1 - \frac{1}{n-f}\right)\frac{\sigma^2}{m}}{(\kappa+1)\left(1 - \frac{1}{n-f}\right)\frac{\sigma^2}{m} + \|\mu_i - \overline{\mu}_\mathcal{C}\|^2 + \kappa\Delta^2}, \tag{6}$$

which approaches 0 as $\|\mu_i - \overline{\mu}_\mathcal{C}\|^2 + \kappa\Delta^2$ grows and $1/(1+\kappa)$ when $\|\mu_i - \overline{\mu}_\mathcal{C}\|^2 + \kappa\Delta^2 \ll \sigma^2/m$.

Note that $\kappa$ can be as small as $\mathcal{O}(f/n)$, hence the above means that, when heterogeneity is limited and with a small enough fraction of adversarial clients, high level of collaborations can be used by each correct client. On the other hand, correct clients need not gain much by collaborating with other clients in highly heterogeneous regimes.

### 2.2 Experimental Validation

To validate our theoretical observations on Byzantine-robust federated mean estimation, we run a series of experiments using simulated datasets sampled from 1-dimensional Gaussian distributions.

---

[3] This estimator need not be optimal a priori. But it represents the state-of-the-art solution for robust FL.

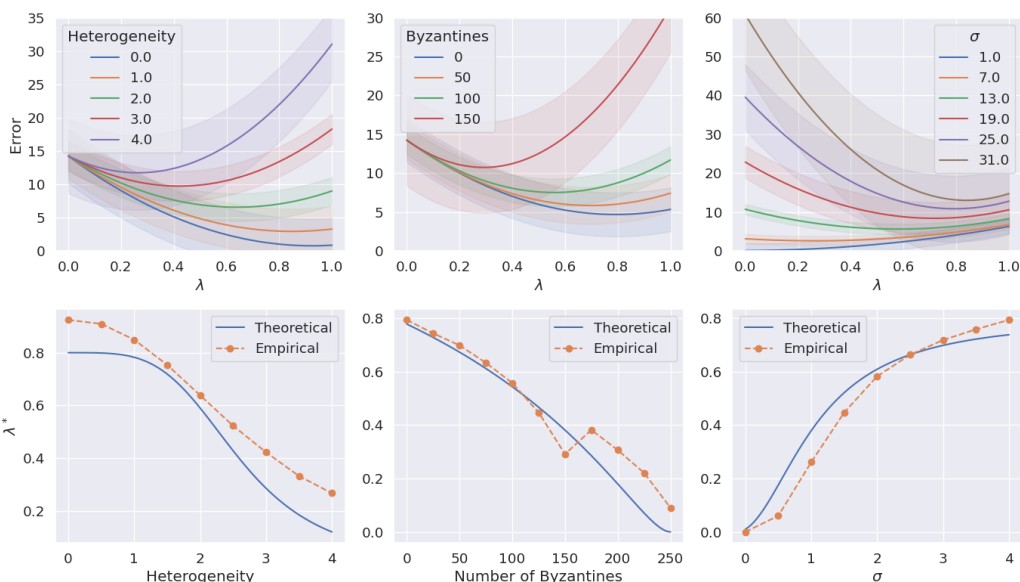

Figure 1: Impact of the heterogeneity ($\sigma_h$), number of Byzantine adversaries $f$ and the task complexity ($\sigma$). (Top) The average error for different values of $\lambda$, computed using 20 random experiments. (Bottom) Comparison of theoretical $\lambda^*$ and empirical minimizer of the error. We fixed the following default values: $n = 600, f = 100, m = 20, \sigma = 15, \sigma_h = 2$

Specifically, for each of the $n - f$ correct clients, we consider that their distribution is such that $\mathcal{D}_i = \mathcal{N}(\mu_i, \sigma^2)$, where the unknown local means $(\mu_i)_{i \in \mathcal{C}}$ have been sampled i.i.d. from a Gaussian distribution $\mathcal{N}(10, \sigma_h^2)$[4], where $\sigma_h$ determines the expected squared distance between the local means of each correct client. Each honest client $i \in \mathcal{C}$ samples $m$ datapoints from $\mathcal{D}_i$ and the error is computed with respect to $\mu_i$. To evaluate the theoretical expression of $\lambda^*$, defined in (6), in this scenario, we replace the distances of the form $(\mu_i - \overline{\mu}_{\mathcal{C}})^2$ by the variance $(1 - 1/n-f)\sigma_h^2$. We present in Figure 1 the average error (squared distance to the true mean) of the estimator defined as per (5) on 20 runs, for different values of $\lambda$. The robust aggregation being used here is an NNM pre-aggregation rule (Allouah et al., 2023) followed by trimmed mean (Yin et al., 2018) and adversarial clients implement the sign-flipping attack. Below we analyze our result presented in the first row of Figure 1.

**Heterogeneity.** First, we study the impact of the heterogeneity level in Figure 1. Doing so, we fix $n = 600, f = 100, m = 20, \sigma = 15$ and vary the level of heterogeneity $\sigma_h \in \{0, 1, 2, 3, 4\}$. We observe that for low levels of heterogeneity, the optimal choice for $\lambda$ is close to 1. In this case, personalization does not really help.

**Byzantine fraction.** Second, we analyze the impact of the fraction of Byzantine adversaries. For Figure 1, we fix $n = 600, m = 20, \sigma = 15, \sigma_h = 2$ and vary $f \in \{0, 50, 100, 150\}$. As per our theoretical analysis, as the Byzantine fraction increases $\kappa$ should increase; hence the correct clients need to rely less and less on the global aggregate. However, simply using a local estimator can be detrimental as showcased by the red curve, which shows the case $f = 150$. Then, by choosing the right collaboration, one reduces the error by $50\%$ compared to the local estimator.

**Task complexity.** Finally, we study the impact of the task complexity, characterized by the quantity $\sigma^2/m$. We fix $n = 600, f = 100, m = 20, \sigma_h = 2$ and vary $\sigma \in \{1, 7, 13, 19, 25, 31\}$. We observe, as expected, that the optimal level of collaboration level of collaboration moves away from 0 as the task gets harder. However, we also observe that, since the fraction of Byzantine clients is non-negligible, the optimal choice of collaboration $\lambda$, even for very large values of $\sigma$, never goes to 1.

**Correspondence between the upper-bound and empirical observations.** The second line of Figure 1 shows a comparison between the optimal theoretical value predicted by our analysis and

---

[4]We use a mean at 10 to break the symmetry of the mean simulation around 0. This is simply to present a case in which the sign-flipping attack is indeed disrupting the federated mean estimation procedure

the empirical minimizer of the error on average.[5] We see that the value we predicted for $\lambda^*$ and the actual empirical best choice for $\lambda$ has very similar trends in all the settings we consider.

# 3 Binary Classification with Adversaries

In this section, we consider the more general problem of binary classification. We study the learning-theoretic setup given in Section 1 with binary output space $\mathcal{Y} := \{0, 1\}$ and $\mathcal{Y}' = [0, 1]$. Throughout, we place ourselves in a hypothesis space $\mathcal{H}$ of finite pseudo-dimension (Vidyasagar, 2003; Mohri et al., 2018), denoted $\mathrm{Pdim}(\mathcal{H})$, which we recall reduces to the VC dimension for the $0 - 1$ loss. We further make the following assumptions on the loss functions. These assumptions are standard in the Byzantine robustness literature see, e.g., (Karimireddy et al., 2022; Allouah et al., 2023).

**Assumption 1** ($L$-smoothness, $\mu$-strong convexity). Each loss function $\mathcal{L}_i, i \in \mathcal{C}$, is $L$-smooth and $\mu$-strongly convex. That is, for all $\theta, \theta' \in \Theta$, we have

$$\frac{\mu}{2}\|\theta - \theta'\|^2 \le \mathcal{L}_i(\theta) - \mathcal{L}_i(\theta') - \langle \nabla \mathcal{L}_i(\theta'), \theta - \theta' \rangle \le \frac{L}{2}\|\theta - \theta'\|^2.$$

**Assumption 2** (Bounded heterogeneity). There exists a real value $G$ such that for all $\theta \in \Theta$, we have

$$\frac{1}{|\mathcal{C}|} \sum_{i \in \mathcal{C}} \|\nabla \mathcal{L}_i(\theta) - \nabla \mathcal{L}_{\mathcal{C}}(\theta)\|^2 \le G^2.$$

We additionally make the following assumption on the parameter set $\Theta$ to ensure that the Euclidean projection on $\Theta$ (denoted $\Pi_\Theta$) is well-defined, unique and that the minimizer of the interpolated loss is not pathological, i.e., is not in the border of $\Theta$.

**Assumption 3.** The parameter set $\Theta$ is compact and convex. Moreover, for every $i \in \mathcal{C}, \lambda \in [0, 1]$, the minimizer of $\mathcal{L}_i^\lambda$ is in the interior of $\Theta$.

Additionally, for our generalization analysis, we make the following assumption on the loss function:

**Assumption 4.** The loss function is bounded in $[0, 1]$.

To conduct an analysis of the effect of collaboration on the generalization performance of correct clients when trying to solve (1), we proceed in two steps. We first evaluate the optimization error that a standard gradient descent algorithm incurs in our setting. Then we bound the generalization gap induced when minimizing (1) and combine the two bounds into our main result in Theorem 1.

## 3.1 Algorithm & Optimization Error

We focus our analysis on a simple personalized variant of the robust distributed gradient descent algorithm, which is the standard algorithm in the Byzantine-robust FL literature and is shown to achieve tight optimization bounds (Allouah et al., 2023). Our variant, presented in Algorithm 1, essentially corresponds to gradient descent on the function $\mathcal{L}_i^\lambda$, but using a robust estimate $R_i^t$ (iteration 7) of the gradient of $\mathcal{L}_{\mathcal{C}}$, which cannot be computed exactly due to Byzantine adversaries. This robust estimate is computed using a robust aggregation $F$, e.g., trimmed mean or median.

**On the execution of the algorithm.** Executing Algorithm 1 in practice implies that each client acts as a local server, and runs an independent federated learning procedure. We further assume that the clients have access to a communication protocol that allows them to broadcast their model to the other clients. This can either be done through the use of the central server or directly through a decentralized communication (i.e., without a server). This algorithm is not communication efficient and we believe that it could be improved in that direction. Nevertheless, it allows us to discuss the information-theoretic aspects of the problem in a simple manner. We first present the optimization error of Algorithm 1 in Lemma 1 below.

**Lemma 1.** *Let assumptions 1, 2, and 3 hold. Consider Algorithm 1 with learning rate $\eta = \frac{1}{2L}$, $\lambda \in [0, 1]$, and assume the aggregation function $F$ to be $(f, \kappa)$-robust. For any $T \ge 1$, we have:*

$$\mathcal{L}_i^\lambda(\theta_i^T) - \mathcal{L}_{i,*}^\lambda \le \frac{5L\lambda^2\kappa G^2}{\mu^2} + \left(1 - \frac{\mu}{2L}\right)^T \frac{L}{\mu}\mathcal{L}_0,$$

*where $\mathcal{L}_0 := \mathcal{L}_i^\lambda(\theta_i^0) - \mathcal{L}_{i,*}^\lambda$ and $\mathcal{L}_{i,*}^\lambda := \min_{\theta \in \Theta} \mathcal{L}_i^\lambda(\theta)$.*

---

[5] $\kappa$ is replaced by $f/n-2f$, following (Allouah et al., 2023)

---
**Algorithm 1** Interpolated Personalized Gradient Descent for client $i \in \mathcal{C}$
---
**Require:** Initialization $\theta_i^0$, aggregation rule $F$, learning rate $\eta$, number of iterations $T$, and collaboration parameter $\lambda$.
1: **for** $t = 1 \ldots T$ **do**
2:     Broadcast $\theta_i^{t-1}$ to all clients
3:     **for** $j = 1 \ldots n, j \neq i$ **do**
4:         Receive $g_{i,j}^t = \nabla \mathcal{L}_j(\theta_i^{t-1})$ from client $j$    ▷ adversarial clients send corrupted gradients
5:     **end for**
6:     Compute local gradient $g_{i,i}^t = \nabla \mathcal{L}_i(\theta_i^{t-1})$
7:     Robustly aggregate $R_i^t = F(g_{i,1}^t, \ldots, g_{i,n}^t)$
8:     Update and project local parameters

$$\theta_i^t = \Pi_\Theta \left( \theta_i^{t-1} - \eta \left( (1-\lambda) g_{i,i}^t + \lambda R_i^t \right) \right)$$

9: **end for**
---

Our optimization error bound recovers the classical GD linear rate when $\lambda = 0$ (local learning), and achieves an asymptotic error in $\mathcal{O}(\kappa G^2)$ when $\lambda = 1$ (full collaboration), which is provably tight under Assumption 2 (Karimireddy et al., 2022). Interestingly, for $\lambda \in (0, 1)$, the asymptotic error smoothly interpolates in $\mathcal{O}(\lambda^2 \kappa G^2)$, which is expected as lesser collaboration limits the influence of the adversary on the optimization error.

### 3.2 Generalization Gap & Effect of Collaboration

Although the presence of adversarial clients impairs the optimization process, one expects a generalization benefit from collaboration if data distributions are similar enough. We formalize this intuition by bounding the generalization gap, of the personalized learning problem (2). We first bound the generalization gap on the interpolated loss $\mathcal{L}_i^\lambda$ using a result adapted from (Blitzer et al., 2007).

**Lemma 2.** *Let Assumption 4 hold. For any $\delta > 0$, $\theta \in \Theta$ and $\lambda \in [0, 1]$, we have with probability at least $1 - \delta$ (over the choice of samples) that*

$$|\mathcal{L}_i^\lambda(\theta) - \mathcal{R}_i^\lambda(\theta)| \leq 2\beta \sqrt{\left( \frac{\left( 1 - \lambda + \frac{\lambda}{n-f} \right)^2}{m} + \frac{\lambda^2}{m(n-f)} \right)},$$

*where $\mathcal{R}_i^\lambda(\theta) := \mathbb{E}[\mathcal{L}_i^\lambda(\theta)], \forall \theta \in \Theta$, and $\beta := \sqrt{\mathrm{Pdim}(\mathcal{H}) \log \left( \frac{em}{\mathrm{Pdim}(\mathcal{H})} \right)} + \sqrt{\log(1/\delta)}$.*

When specializing the result of Lemma 2 to $\lambda = 0$ and $\lambda = 1$, we recover the standard generalization gaps $\mathcal{O}\left( \sqrt{\mathrm{Pdim}(\mathcal{H})/m} \right)$ and $\mathcal{O}\left( \sqrt{\mathrm{Pdim}(\mathcal{H})/m(n-f)} \right)$ for local learning and federated learning (with correct clients only), respectively. However, in addition to the generalization gap bounded in Lemma 2, we need to bound the gap between the interpolated risk (3) and the original local risk (1). In fact, these two objectives are statistically different when there is data heterogeneity among clients since the interpolated risk (3) involves the average of local loss functions. To quantify this difference, we leverage tools from domain adaptation theory (Ben-David et al., 2010) and consider a function $\Phi$ measuring the discrepancy between two statistical distributions. Formally, we require for every $i \in \mathcal{C}$,

$$|\mathcal{R}_i(\theta) - \mathcal{R}_\mathcal{C}(\theta)| \leq \Phi(\mathcal{D}_i, \mathcal{D}_\mathcal{C}), \forall \theta \in \Theta. \tag{7}$$

For example, for the case of 0-1 loss in binary classification, for any two distributions $\mathcal{D}_1$ and $\mathcal{D}_2$, (Blitzer et al., 2007) propose the discrepancy measure to be $\Phi(\mathcal{D}_1, \mathcal{D}_2) = d_{\mathcal{H}\Delta\mathcal{H}}(\mathcal{D}_1, \mathcal{D}_2) := 2 \sup_{h \in \mathcal{H}} |\mathbb{P}_{\mathcal{D}_1}(I(h)) - \mathbb{P}_{\mathcal{D}_2}(I(h))|$, where $I(h) := \{x \in \mathcal{X} : h(x) = 1\}$. For more general losses, we can use hypothesis space-dependent Integral Probability Metrics (Sriperumbudur et al., 2009; Bao et al., 2023), which we include in Appendix B.2. We now combine our bounds on the optimization error and generalization gap into Theorem 1 below.

**Theorem 1.** *Let assumptions 1, 2, 3, and 4 hold, and let $\Phi$ be a function such that (7) holds. Consider Algorithm 1 with learning rate $\eta = \frac{1}{2L}$, $\lambda \in [0, 1]$, and assume the aggregation function $F$ to be*

$(f, \kappa)$-*robust. Then, for any* $\delta > 0, T \geq 1$, *we have with probability at least* $1 - \delta$ *(over the choice of samples) that*

$$\mathcal{R}_i(\theta_i^T) - \mathcal{R}_i(\theta_i^*) \leq \left(1 - \frac{\mu}{2L}\right)^T \frac{L}{\mu} \mathcal{L}_0 + \frac{5L\lambda^2 \kappa G2}{\mu^2} + 2\lambda \Phi(\mathcal{D}_i, \mathcal{D}_\mathcal{C})$$

$$+ 4\beta \sqrt{\frac{\left(1 - \lambda + \frac{\lambda}{n-f}\right)^2}{m} + \frac{\lambda^2}{m(n-f)}}, \tag{8}$$

*where* $\beta := \sqrt{\mathrm{Pdim}(\mathcal{H}) \log\left(\frac{em}{\mathrm{Pdim}(\mathcal{H})}\right)} + \sqrt{\log(1/\delta)}$ *and* $\mathcal{L}_{i,*}^\lambda := \min_{\theta \in \Theta} \mathcal{L}_i^\lambda(\theta)$.

**Interpretation.** The bound in Theorem 1 features a trade-off between *data heterogeneity*, *sample size*, and *model complexity*. Indeed, we minimize the bound in Theorem 1 and obtain the following closed-form approximation by ignoring constant and logarithmic terms, and assuming that $n, T \gg 1$ (see Appendix C for details)

$$\lambda^* \approx \Pi_{[0,1]} \left( \frac{\sqrt{\frac{\mathrm{Pdim}(\mathcal{H})}{m}} - \Phi(\mathcal{D}_i, \mathcal{D}_\mathcal{C})}{\frac{f}{n} G^2} \right), \tag{9}$$

where $\Pi_{[0,1]}$ denotes the projection over $[0, 1]$. This expression suggests that the optimal collaboration parameter decreases with data heterogeneity. Indeed, if $\Phi(\mathcal{D}_i, \mathcal{D}_\mathcal{C}) > \sqrt{\mathrm{Pdim}(\mathcal{H})/m}$, then $\lambda^* = 0$ and local learning, as expected, is optimal. Otherwise, the optimal collaboration parameter can be non-zero. This validates the fact that correct clients cannot improve their local generalization when their local distributions are too dissimilar compared to the complexity of the hypothesis class and the number of data points each of them has. Furthermore, even if the heterogeneity $\Phi(\mathcal{D}_i, \mathcal{D}_\mathcal{C})$ sufficiently small compared to $\sqrt{\mathrm{Pdim}(\mathcal{H})/m}$, the level of collaboration must be re-evaluated according to $\frac{f}{n} G^2$; hence depending on both the fraction of adversaries and the dissimilarity of the gradient between correct clients.

### 3.3 Experimental Validation

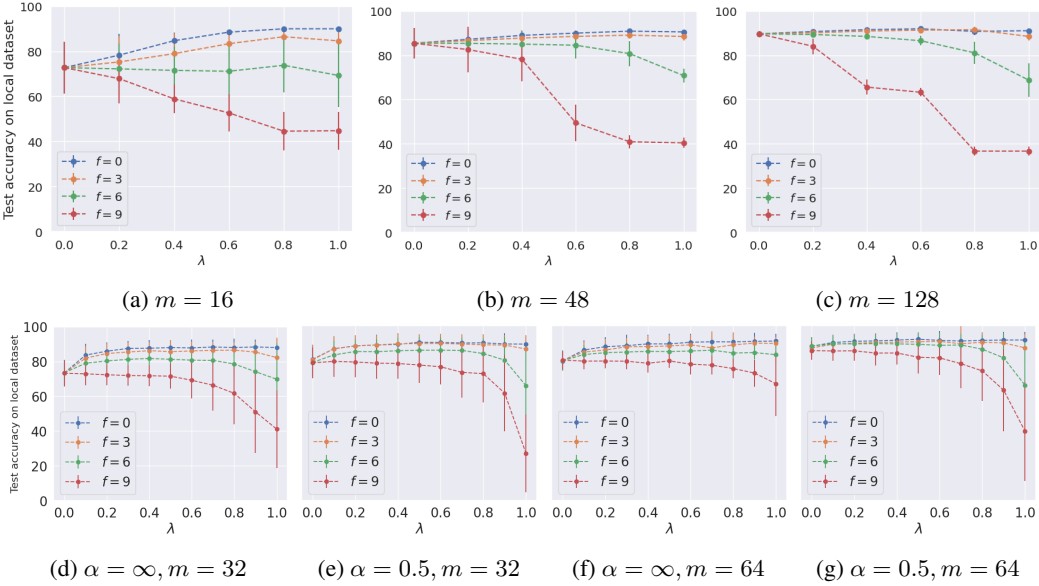

(a) $m = 16$     (b) $m = 48$     (c) $m = 128$

(d) $\alpha = \infty, m = 32$   (e) $\alpha = 0.5, m = 32$   (f) $\alpha = \infty, m = 64$   (g) $\alpha = 0.5, m = 64$

Figure 2: Effect of adversarial fraction and heterogeneity and local sample size. (Top) Phishing dataset with logistic regression with $n = 20, \alpha = 3$. (Bottom) MNIST with a Convolutional Neural Network (details in Appendix D) $n = 20$. $\alpha = \infty$ refers to the homogeneous setting.

**Setup.** We empirically investigate the impact of Byzantine adversaries on the generalization performance, in our personalization framework, using the Phishing dataset (Chiew et al., 2019) and

the MNIST dataset (LeCun and Cortes, 2010). Other experiments are included in Appendix D. For the different experiments, we use the state-of-the-art defense under data heterogeneity: NNM pre-aggregation (Allouah et al., 2023) rule followed by trimmed mean as robust aggregation (Yin et al., 2018). We simulate the Byzantine attack with the Sign Flipping attack (Li et al., 2020). Throughout the experiments, we fix $n = 20$, and only vary the local dataset size $m$, the fraction of Byzantine adversaries $f$, and the heterogeneity level $\alpha$ which we will explain in the next paragraph. The clients execute Algorithm 1 for $T$ iterations which depends on the dataset being used. Figure 2, as well as the figures in Appendix D, show the average results and error bars for 5 random runs.

To simulate heterogeneity, we use a Dirichlet distribution (Hsu et al., 2019) to generate datasets with unbalanced class fractions for each client. The parameter $\alpha$ determines the heterogeneity degree (the smaller the bigger the heterogeneity). We subsequently sample the test datasets using the same class distribution as the train datasets for each client, and we evaluate the trained models on these local datasets. We defer the implementation details to Appendix D.

Figure 2, as well as Figure 6 from Appendix D, show the final local test accuracy performance as a function of the degree of collaboration ($\lambda$) for different values of $f, m$ and $\alpha$. These experimental results shed light on the impact of these different factors and allow us to confirm the main insights of our theory.

**Full collaboration can be suboptimal.** This can be gleaned from settings where the adversarial fraction is substantial (over 6 adversarial clients, e.g. green and red curves in Figure 2), when the heterogeneity is large (small values of $\alpha$, e.g. blue and orange curves in Figure 6 in Appendix D) and when the local dataset is large enough (e.g. Figure 2b and Figure 2c). In situations where the adversarial fraction is critical ($f = 9$, corresponding to red curves in Figure 2 ), robust federated learning completely fails, achieving accuracy scores under $50\%$ on the local test datasets. Figure 5 in Appendix D further illustrates this point, showing that even local learning can be better than state-of-the-art robust Federated Learning in these circumstances.

**Fine-tuning the collaboration helps get the best of both worlds.** Figure 2(Bottom) shows that in data scarcity scenarios, for moderate values of the number of adversarial clients ($f = 3$ or $f = 6$), using a collaboration degree strictly between $0$ and $1$ yields better accuracy on the local test dataset. The same effect also appears in Figure 2(Top) for $f = 3$ and Figure 4 in Appendix D. Additionally, even for extreme adversarial fractions ($f = 9$ in our experiments), the gain from decreasing the collaboration degree can be substantial compared to simply using Robust Federated Learning methods. This suggests that fine-tuning the collaboration degree can help dampen the detrimental effects of adversarial clients.

# 4   Conclusion & Future Work

In this paper, we have taken a first step towards understanding how fine-tuning personalization in FL can mitigate the impact of adversarial clients. The results we obtain suggest that the use of personalization can improve the performance of FL algorithms in the presence of adversarial clients, but also that the level of collaboration needs to be chosen carefully. Our theoretical analysis accounts for the necessity to strike a balance between the optimization error, which is impacted by adversarial clients, and the generalization gap, which is likely to benefit from a larger pool of data. We identify the main factors that impact the local performance, namely data heterogeneity, the fraction of adversarial clients, and data scarcity. This work could be extended in several directions. Firstly, as explained in Section 3, the study of Algorithm 1 primarily serves our understanding of the robust interpolated optimization problem in (3) from an information-theoretic perspective. However, it may not be efficient enough to cope with high-dimensional learning tasks involving a number of clients. An interesting future direction could be to investigate alternative algorithms with lower communication and computational costs, which would be able to handle these high-dimensional problems. Second, our study focuses on a personalization strategy that interpolates between client loss functions. Although this strategy is simple to interpret and allows us to derive optimization and generalization bounds, it is only one of many possible personalized FL schemes. Another interesting avenue would be to investigate whether the insights we drew from our interpolated problem could be derived in a similar fashion from other personalization frameworks such as regularization, layer specialization, or clustering.

## Acknowledgments and Disclosure of Funding

This work was supported in part by the Swiss National Science Foundation under Grant N°20CH21_218778, TruBrain and by SNSF grant 200021_200477. The authors are thankful to the anonymous reviewers for their constructive comments.

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

# A Related Work

**Personalized federated learning.** As federated learning methods are typically challenged by data heterogeneity (Kairouz et al., 2021), many personalization methods have recently been proposed to learn different models tailored for each client. A first category of personalized learning algorithms is based upon local fine-tuning (Fallah et al., 2020; Dinh et al., 2020), where the goal is to collaboratively find a good global model from which local fine-tuning can result in an improved local performance. A second category of personalized algorithms is based upon regularization between local models (Li et al., 2021; Kundu et al., 2022), i.e., adding a regularization term penalizing a distance between local client models. Another line of research in personalized federated learning considers partial personalization where the parameters are split into globally shared and individual local parameters (Bietti et al., 2022; Mishchenko et al., 2023), this, however, creates the additional challenge of choosing this split. Personalization can also be implemented through clustering methods, where participants selectively collaborate with a smaller subset of clients to limit their exposure to heterogeneity (Ghosh et al., 2020; Werner et al., 2023; Bao et al., 2023). (Mansour et al., 2020) proposed and analyzed three different interpolation methods to formalize personalization, including the one we consider in this paper, which they refer to as *data interpolation*. Although they discuss the generalization properties of this model, they do not address Byzantine robustness.

**Personalization for Byzantine robustness.** A few works in the personalized learning literature suggest that personalization can improve the robustness to Byzantine participants (Li et al., 2021; Kundu et al., 2022; Mishchenko et al., 2023). These works however only present a limited theoretical analysis in the general Byzantine context. For instance, contrary to our work, the theoretical results in (Li et al., 2021) only cover linear regression and random Gaussian noise attacks. Moreover, the results of (Mishchenko et al., 2023) concerning Byzantine robustness make very strong assumptions on the heterogeneity of loss functions, essentially assuming that they share a common minimum, which largely mitigates the effect of Byzantine adversaries on the optimization error (Allouah et al., 2024). Both (Kundu et al., 2022; Mishchenko et al., 2023) only analyze the optimization error and do not study the generalization error, which we showcase as an important part of the trade-off.

# B Definitions

## B.1 Strong Convexity

A close, but weaker version of strong convexity is the Polyak-Lojasiewicz condition defined below:

**Definition 2** (Polyak-Lojasiewicz (PL)). A function $\mathcal{L} : \mathbb{R}^d \to \mathbb{R}$ is said to satisfy Polyak-Lojasiewicz with parameter $\mu$, noted $\mu$-PL if $\forall x \in \mathbb{R}^d$:

$$\mathcal{L}(x) - \inf_{\mathbb{R}^d} \mathcal{L} \leq \frac{1}{2\mu} \|\nabla \mathcal{L}(x)\|^2 \tag{10}$$

If a function $\mathcal{L}$ is $\mu$-strongly convex, it satisfies $\mu$-PL condition. Indeed a $\mu$-strongly convex function $\mathcal{L}$ satisfies:

$$\frac{\mu}{2}\|x - y\|^2 + \langle \nabla \mathcal{L}(y),\, x - y \rangle \leq \mathcal{L}(x) - \mathcal{L}(y). \tag{11}$$

Thus, $\mu$-PL can be obtained from the above by minimizing each side with respect to $x$.

## B.2 Discrepancy

For the 0-1 loss, let $d_{\mathcal{H}}$ be defined as follows :

$$d_{\mathcal{H}}(D, D') = 2 \sup_{h \in \mathcal{H}} |\mathbb{P}_D(I(h)) - \mathbb{P}_{D'}(I(h))|, \tag{12}$$

where $I(h) = \{x : h(x) = 1\}$.

Define the symmetric difference hypothesis space : $\mathcal{H}\Delta\mathcal{H} := \{h(x) \oplus h'(x)\} : h, h' \in \mathcal{H}$. Then following (Blitzer et al., 2007), $d_{\mathcal{H}\Delta\mathcal{H}}$ satisfies (7).

For general losses, we define the following discrepancy:

$$\phi(D_1, D_2) = \max_{h \in \mathcal{H}} |\mathbb{E}_{D_1} \mathcal{L}(h(x), y) - \mathbb{E}_{D_2} \mathcal{L}(h(x), y)|$$

# C Proofs

## C.1 Proof of Mean Point Estimation

**Proposition 1.** *Consider the mean estimation setting described. For any $i \in \mathcal{C}$, let $y_i^\lambda$ be as defined in (5) with an aggregation rule $F$ that satisfies $(f, \kappa)$-robustness. Then the following holds true:*

$$\mathbb{E}\left[\|y_i^\lambda - \mu_i\|^2\right] \leq 3\left(1 - \frac{1}{n-f}\right)\frac{\sigma^2 \Gamma(\lambda, \kappa)}{m} + 3\lambda^2(\|\mu_i - \overline{\mu}_{\mathcal{C}}\|^2 + \kappa\Delta^2).$$

*with $\overline{\mu}_{\mathcal{C}} := \frac{1}{n-f}\sum_{i \in \mathcal{C}}\mu_i$, $\Delta^2 := \frac{1}{n-f}\sum_{j \in \mathcal{C}}\|\mu_j - \overline{\mu}_{\mathcal{C}}\|^2$, and $\Gamma(\lambda, \kappa) := \lambda^2(\kappa + 1) - 2\lambda + \frac{n-f}{n-f-1}$.*

*Proof.* We can bound the error as follows:

$$\mathbb{E}\left[\|y_i^\lambda - \mu_i\|^2\right] = \mathbb{E}\left[\|(1-\lambda)(\widehat{y_i} - \mu_i) + \lambda(\widehat{y_{\mathcal{C}}} - \mu_{\mathcal{C}}) + \lambda(\mu_{\mathcal{C}} - \mu_i) + \lambda(R - \widehat{y_{\mathcal{C}}})\|^2\right]$$

$$\leq 3\mathbb{E}\left[\|(1-\lambda)(\widehat{y_i} - \mu_i) + \lambda(\widehat{y_{\mathcal{C}}} - \mu_{\mathcal{C}})\|^2\right] + 3\lambda^2 \mathbb{E}\left[\|R - \widehat{y_{\mathcal{C}}}\|^2\right] + 3\lambda^2\|\mu_{\mathcal{C}} - \mu_i\|^2. \tag{13}$$

where, for the second line we use the triangle inequality and the fact that $(a+b+c)^2 \leq 3(a^2+b^2+c^2)$, for all $a, b, c \geq 0$.

**Upper-bounding the first term.** The first term can be simplified as follows:

$$\mathbb{E}\left[\|(1-\lambda)(\widehat{y_i} - \mu_i) + \lambda(\widehat{y_{\mathcal{C}}} - \mu_{\mathcal{C}})\|^2\right] = \mathbb{E}\left[\left\|\left(1 - \lambda + \frac{\lambda}{|\mathcal{C}|}\right)(\widehat{y_i} - \mu_i) + \frac{\lambda}{|\mathcal{C}|}\sum_{j \neq i}(\widehat{y_j} - \mu_j)\right\|^2\right]$$

Because the data points are sampled independently for each client, we have that

$$\mathbb{E}\left[\|(1-\lambda)(\widehat{y_i} - \mu_i) + \lambda(\widehat{y_{\mathcal{C}}} - \mu_{\mathcal{C}})\|^2\right] = \left(1 - \lambda + \frac{\lambda}{|\mathcal{C}|}\right)^2 \mathbb{E}\left[\|\widehat{y_i} - \mu_i\|^2\right] + \frac{\lambda^2}{|\mathcal{C}|^2}\mathbb{E}\left[\left\|\sum_{j \neq i}(\widehat{y_j} - \mu_j)\right\|^2\right].$$

Because the data points are sampled i.i.d. each client local distributing and independently for each client, we also have

$$\mathbb{E}\left[\|(1-\lambda)(\widehat{y_i} - \mu_i) + \lambda(\widehat{y_{\mathcal{C}}} - \mu_{\mathcal{C}})\|^2\right] = \left(1 - \lambda + \frac{\lambda}{|\mathcal{C}|}\right)^2 \mathbb{E}\left[\|\widehat{y_i} - \mu_i\|^2\right] + \frac{\lambda^2}{|\mathcal{C}|^2}\sum_{j \neq i}\mathbb{E}\left[\|(\widehat{y_j} - \mu_j)\|^2\right]$$

$$= \left(1 - \lambda + \frac{\lambda}{|\mathcal{C}|}\right)^2 \frac{1}{m}\mathbb{E}_{y \sim \mathcal{D}_{i|\mathcal{Y}}}\left[\|y - \mu_i\|^2\right] + \frac{\lambda^2}{|\mathcal{C}|^2}\sum_{j \neq i}\frac{1}{m}\mathbb{E}_{y \sim \mathcal{D}_{j|\mathcal{Y}}}\left[\|y - \mu_j\|^2\right]$$

$$= \left(\left(1 - \lambda + \frac{\lambda}{|\mathcal{C}|}\right)^2 + \frac{\lambda^2(|\mathcal{C}| - 1)}{|\mathcal{C}|^2}\right)\frac{\sigma^2}{m}.$$

Where the last line comes from the fact that $\mathbb{E}_{y \sim \mathcal{D}_{i|\mathcal{Y}}}\left[\|y - \mu_i\|^2\right] = \sigma_i = \sigma^2$ for all $i \in [n]$.

**Upper-bounding the second term.** The second term can be controlled by the $(f, \kappa)$-robustness property of $F$ as follows

$$\lambda^2 \mathbb{E}\left[\|R - \widehat{y_{\mathcal{C}}}\|^2\right] = \lambda^2 \mathbb{E}\left[\|F(\widehat{y_1}, \ldots, \widehat{y_n}) - \widehat{y_{\mathcal{C}}}\|^2\right] \leq \lambda^2 \frac{\kappa}{|\mathcal{C}|} \sum_{i \in \mathcal{C}} \mathbb{E}\left[\|\widehat{y_i} - \widehat{y_{\mathcal{C}}}\|^2\right]$$

Then using similar arguments as for the first term we have

$$
\begin{aligned}
\mathbb{E}\left[\|\widehat{y_i} - \widehat{y_{\mathcal{C}}}\|^2\right] &= \mathbb{E}\left[\|\widehat{y_i} - \mu_i + (\mu_i - \mu_{\mathcal{C}}) + \mu_{\mathcal{C}} - \widehat{y_{\mathcal{C}}}\|^2\right] \\
&= \mathbb{E}\left[\|\widehat{y_i} - \mu_i + \mu_{\mathcal{C}} - \widehat{y_{\mathcal{C}}}\|^2\right] + \|\mu_i - \mu_{\mathcal{C}}\|^2 \\
&= \mathbb{E}\left[\left\|\left(1 - \frac{1}{|\mathcal{C}|}\right)(\widehat{y_i} - \mu_i) + \frac{1}{|\mathcal{C}|}\sum_{j \neq i}(\mu_j - \widehat{y_j})\right\|^2\right] + \|\mu_i - \mu_{\mathcal{C}}\|^2 \\
&= \mathbb{E}\left[\left\|\left(1 - \frac{1}{|\mathcal{C}|}\right)(\widehat{y_i} - \mu_i)\right\|^2\right] + \mathbb{E}\left[\left\|\frac{1}{|\mathcal{C}|}\sum_{j \neq i}(\mu_j - \widehat{y_j})\right\|^2\right] + \|\mu_i - \mu_{\mathcal{C}}\|^2 \\
&= \left(1 - \frac{1}{|\mathcal{C}|}\right)^2 \frac{\sigma^2}{m} + \frac{(|\mathcal{C}| - 1)}{|\mathcal{C}|^2}\frac{\sigma^2}{m} + \|\mu_i - \mu_{\mathcal{C}}\|^2 \\
&= \left(\left(1 - \frac{1}{|\mathcal{C}|}\right)^2 + \frac{(|\mathcal{C}| - 1)}{|\mathcal{C}|^2}\right)\frac{\sigma^2}{m} + \|\mu_i - \mu_{\mathcal{C}}\|^2
\end{aligned}
$$

Substituting this in the above we get

$$
\begin{aligned}
\lambda^2 \mathbb{E}\left[\|R - \widehat{y_{\mathcal{C}}}\|^2\right] &\leq \kappa\lambda^2\left(\left(1 - \frac{1}{|\mathcal{C}|}\right)^2 + \frac{(|\mathcal{C}| - 1)}{|\mathcal{C}|^2}\right)\frac{\sigma^2}{m} + \frac{\kappa\lambda^2}{|\mathcal{C}|}\sum_{j \in \mathcal{C}}\|\mu_j - \mu_{\mathcal{C}}\|^2 \\
&\leq \frac{|\mathcal{C}| - 1}{|\mathcal{C}|}\kappa\lambda^2\frac{\sigma^2}{m} + \frac{\kappa\lambda^2}{|\mathcal{C}|}\sum_{j \in \mathcal{C}}\|\mu_j - \mu_{\mathcal{C}}\|^2
\end{aligned}
$$

**Conclusion.** By substituting the above in (13), we get the following.

$$
\begin{aligned}
\mathbb{E}[\|y_i^\lambda - \mu_i\|^2] \leq 3\Bigg( &\frac{|\mathcal{C}| - 1}{|\mathcal{C}|}\kappa\lambda^2 + \left(1 - \lambda + \frac{\lambda}{|\mathcal{C}|}\right)^2 + \frac{\lambda^2(|\mathcal{C}| - 1)}{|\mathcal{C}|^2}\Bigg)\frac{\sigma^2}{m} \\
&+ 3\lambda^2\|\mu_i - \mu_{\mathcal{C}}\|^2 + 3\frac{\kappa\lambda^2}{|\mathcal{C}|}\sum_{j \in \mathcal{C}}\|\mu_j - \mu_{\mathcal{C}}\|^2
\end{aligned}
\tag{14}
$$

Using the notation $\Delta^2 = \frac{1}{|\mathcal{C}|}\sum_{j \in \mathcal{C}}\|\mu_j - \mu_{\mathcal{C}}\|^2$, we get

$$\mathbb{E}[\|y_i^\lambda - \mu_i\|^2] \leq 3\left(\frac{|\mathcal{C}| - 1}{|\mathcal{C}|}\lambda^2(\kappa + 1) + 1 - 2\frac{|\mathcal{C}| - 1}{|\mathcal{C}|}\lambda\right)\frac{\sigma^2}{m} + 3\lambda^2(\|\mu_i - \overline{\mu}_{\mathcal{C}}\|^2 + \kappa\Delta^2).$$

$\square$

### C.2 Proof of Lemma 1

**Lemma 1.** *Let assumptions 1, 2, and 3 hold. Consider Algorithm 1 with learning rate $\eta = \frac{1}{2L}$, $\lambda \in [0, 1]$, and assume the aggregation function $F$ to be $(f, \kappa)$-robust. For any $T \geq 1$, we have:*

$$\mathcal{L}_i^\lambda(\theta_i^T) - \mathcal{L}_{i,*}^\lambda \leq \frac{5L\lambda^2\kappa G^2}{\mu^2} + \left(1 - \frac{\mu}{2L}\right)^T \frac{L}{\mu}\mathcal{L}_0,$$

*where $\mathcal{L}_0 := \mathcal{L}_i^\lambda(\theta_i^0) - \mathcal{L}_{i,*}^\lambda$ and $\mathcal{L}_{i,*}^\lambda := \min_{\theta \in \Theta} \mathcal{L}_i^\lambda(\theta)$.*

*Proof.* Let assumptions 1, 2, and 3 hold. First, we note that $\mathcal{L}_i^\lambda$ is $L$-smooth (and $\mu$-strongly convex) as a convex combination of $\mathcal{L}_i$ and $\mathcal{L}_\mathcal{C}$, which are both $L$-smooth (and $\mu$-strongly convex) by Assumption 1. In the remainder of this proof, we denote by $\widehat{\theta}_i^\lambda$ the minimizer of $\mathcal{L}_i^\lambda$ on $\Theta$, i.e., $\mathcal{L}_i^\lambda(\widehat{\theta}_i^\lambda) = \mathcal{L}_{i,*}^\lambda$. Recall also that, by Assumption 3, $\widehat{\theta}_i^\lambda$ is also a critical point of $\mathcal{L}_i^\lambda$.

Let $t \in \{0, \dots, T-1\}$ and denote $\xi_i^t := R_i^t - \nabla\mathcal{L}_\mathcal{C}(\theta_i^t)$. We rewrite the update step as follows:
$$\theta_i^{t+1} = \Pi_\Theta\left(\theta_i^t - \eta\left((1-\lambda)\nabla\mathcal{L}_i(\theta_i^t) + \lambda R_i^t\right)\right) = \Pi_\Theta\left(\theta_i^t - \eta\left(\nabla\mathcal{L}_i^\lambda(\theta_i^t) + \lambda\left(R_i^t - \nabla\mathcal{L}_\mathcal{C}(\theta_i^t)\right)\right)\right).$$

Since the Euclidean projection operator $\Pi_\Theta$ is non-expansive, and $\widehat{\theta}_i^\lambda \in \Theta$ by definition, we have
$$\|\theta_i^{t+1} - \widehat{\theta}_i^\lambda\|^2 \le \|\theta_i^t - \widehat{\theta}_i^\lambda - \eta\left(\nabla\mathcal{L}_i^\lambda(\theta_i^t) + \lambda\xi_i^t\right)\|^2. \tag{15}$$
By developing the right-hand side (15) and using Jensen's inequality, we get
$$\|\theta_i^{t+1} - \widehat{\theta}_i^\lambda\|^2 \le \|\theta_i^t - \widehat{\theta}_i^\lambda\|^2 - 2\eta\left\langle\theta_i^t - \widehat{\theta}_i^\lambda, \nabla\mathcal{L}_i^\lambda(\theta_i^t)\right\rangle - 2\eta\left\langle\theta_i^t - \widehat{\theta}_i^\lambda, \lambda\xi_i^t\right\rangle + \eta^2\|\nabla\mathcal{L}_i^\lambda(\theta_i^t) + \lambda\xi_i^t\|^2$$
$$\le \|\theta_i^t - \widehat{\theta}_i^\lambda\|^2 - 2\eta\left\langle\theta_i^t - \widehat{\theta}_i^\lambda, \nabla\mathcal{L}_i^\lambda(\theta_i^t)\right\rangle - 2\eta\left\langle\theta_i^t - \widehat{\theta}_i^\lambda, \lambda\xi_i^t\right\rangle + 2\eta^2\|\nabla\mathcal{L}_i^\lambda(\theta_i^t)\|^2 + 2\eta^2\lambda^2\|\xi_i^t\|^2.$$

Furthermore, by Assumption 3, we know that $\widehat{\theta}_i^\lambda$ is a critical point of $\mathcal{L}_i^\lambda$ (Assumption 1), hence using the $L$-smooth of $\mathcal{L}_i^\lambda$ we have $\|\nabla\mathcal{L}_i^\lambda(\theta_i^t)\|^2 \le 2L(\mathcal{L}_i^\lambda(\theta_i^t) - \mathcal{L}_{i,*}^\lambda)$. Moreover, since $\mathcal{L}_i^\lambda$ is $\mu$-strongly convex (Assumption 1), we have $\left\langle\theta_i^t - \widehat{\theta}_i^\lambda, \nabla\mathcal{L}_i^\lambda(\theta_i^t)\right\rangle \ge \mathcal{L}_i^\lambda(\theta_i^t) - \mathcal{L}_{i,*}^\lambda + \frac{\mu}{2}\|\theta_i^t - \widehat{\theta}_i^\lambda\|^2$. Substituting these in the above yields
$$\|\theta_i^{t+1} - \widehat{\theta}_i^\lambda\|^2 \le \|\theta_i^t - \widehat{\theta}_i^\lambda\|^2 - 2\eta(\mathcal{L}_i^\lambda(\theta_i^t) - \mathcal{L}_{i,*}^\lambda) - \eta\mu\|\theta_i^t - \widehat{\theta}_i^\lambda\|^2$$
$$- 2\eta\left\langle\theta_i^t - \widehat{\theta}_i^\lambda, \lambda\xi_i^t\right\rangle + 4\eta^2 L(\mathcal{L}_i^\lambda(\theta_i^t) - \mathcal{L}_{i,*}^\lambda) + 2\eta^2\lambda^2\|\xi_i^t\|^2.$$

Given that $\eta = \frac{1}{2L}$, we can simplify this inequality as follows,
$$\|\theta_i^{t+1} - \widehat{\theta}_i^\lambda\|^2 \le \|\theta_i^t - \widehat{\theta}_i^\lambda\|^2 - \eta\mu\|\theta_i^t - \widehat{\theta}_i^\lambda\|^2 - 2\eta\left\langle\theta_i^t - \widehat{\theta}_i^\lambda, \lambda\xi_i^t\right\rangle + 2\eta^2\lambda^2\|\xi_i^t\|^2.$$

We now use Young's inequality on scalar product to get $-\left\langle\theta_i^t - \widehat{\theta}_i^\lambda, \lambda\xi_i^t\right\rangle \le \frac{\mu}{4}\|\theta_i^t - \widehat{\theta}_i^\lambda\|^2 + \frac{4}{\mu}\lambda^2\|\xi_t^t\|^2$. Substituting these in the above yields
$$\|\theta_i^{t+1} - \widehat{\theta}_i^\lambda\|^2 \le \|\theta_i^t - \widehat{\theta}_i^\lambda\|^2 - \eta\mu\|\theta_i^t - \widehat{\theta}_i^\lambda\|^2 + \eta\frac{\mu}{2}\|\theta_i^t - \widehat{\theta}_i^\lambda\|^2 + \eta\frac{8}{\mu}\lambda^2\|\xi_t^t\|^2 + 2\eta^2\lambda^2\|\xi_i^t\|^2$$
$$\le \left(1 - \frac{\eta\mu}{2}\right)\|\theta_i^t - \widehat{\theta}_i^\lambda\|^2 + \lambda^2\left(\eta\frac{8}{\mu} + 2\eta^2\right)\|\xi_i^t\|^2.$$

Substituting $\eta = \frac{1}{2L}$ in the above, and using the fact that $L \ge \mu$, we obtain
$$\|\theta_i^{t+1} - \widehat{\theta}_i^\lambda\|^2 \le \left(1 - \frac{\mu}{2L}\right)\|\theta_i^t - \widehat{\theta}_i^\lambda\|^2 + \lambda^2\left(\frac{4}{\mu L} + \frac{1}{2L^2}\right)\|\xi_i^t\|^2 \le \left(1 - \frac{\mu}{2L}\right)\|\theta_i^t - \widehat{\theta}_i^\lambda\|^2 + \lambda^2\frac{5}{\mu L}\|\xi_i^t\|^2.$$
By $(f, \kappa)$-robustness of $F$, and Assumption 2, we have $\|\xi_i^t\|^2 \le \frac{\kappa}{|\mathcal{C}|}\sum_{i\in\mathcal{C}}\|\nabla\mathcal{L}_i(\theta_i^t) - \nabla\mathcal{L}_\mathcal{C}(\theta_i^t)\|^2 \le \kappa G^2$. Hence, we get
$$\|\theta_i^{t+1} - \widehat{\theta}_i^\lambda\|^2 \le \left(1 - \frac{\mu}{2L}\right)\|\theta_i^t - \widehat{\theta}_i^\lambda\|^2 + \lambda^2\frac{5}{\mu L}\kappa G^2.$$
Recursively using the above, we get
$$\|\theta_i^{t+1} - \widehat{\theta}_i^\lambda\|^2 \le \frac{5\lambda^2\kappa G^2}{\mu L}\sum_{k=0}^{t}\left(1 - \frac{\mu}{2L}\right)^k + \left(1 - \frac{\mu}{2L}\right)^{t+1}\|\theta_i^0 - \widehat{\theta}_i^\lambda\|^2 \tag{16}$$
$$\le \frac{10\lambda^2\kappa G^2}{\mu^2} + \left(1 - \frac{\mu}{2L}\right)^{t+1}\|\theta_i^0 - \widehat{\theta}_i^\lambda\|^2. \tag{17}$$

Combining Assumption 1 and Assumption 3, we easily get that $\frac{\mu}{2}\|\theta - \widehat{\theta}_i^\lambda\|^2 \le \mathcal{L}_i^\lambda(\theta) - \mathcal{L}_{i,*}^\lambda \le \frac{L}{2}\|\theta - \widehat{\theta}_i^\lambda\|^2$ for all $\theta \in \Theta$. Using this, and specializing (16) for $t = T - 1$, we have
$$\mathcal{L}_i^\lambda(\theta_i^T) - \mathcal{L}_{i,*}^\lambda \le \frac{5L\lambda^2\kappa G^2}{\mu^2} + \left(1 - \frac{\mu}{2L}\right)^T\frac{L}{\mu}(\mathcal{L}_i^\lambda(\theta_i^0) - \mathcal{L}_{i,*}^\lambda).$$

$\square$

## C.3 Proof of Lemma 2

We start by reminding Hoeffding inequality:

**Lemma 3** (Hoeffding inequality). *If $X_1, \ldots, X_N$ are real-valued independent random variables, each almost surely belonging to interval $[a_i, b_i]$, then the sum $S_N = \sum\limits_{i \in [N]} X_i$ satisfies*

$$\mathbb{P}\left[|S_N - \mathbb{E}[S_N]| \geq \varepsilon\right] \leq 2 \exp\left(\frac{-2\varepsilon^2}{\sum\limits_{i \in [N]} (b_i - a_i)^2}\right) \tag{18}$$

Back to our problem:

**Lemma 2.** *Let Assumption 4 hold. For any $\delta > 0$, $\theta \in \Theta$ and $\lambda \in [0, 1]$, we have with probability at least $1 - \delta$ (over the choice of samples) that*

$$|\mathcal{L}_i^\lambda(\theta) - \mathcal{R}_i^\lambda(\theta)| \leq 2\beta \sqrt{\left(\frac{\left(1 - \lambda + \frac{\lambda}{n-f}\right)^2}{m} + \frac{\lambda^2}{m(n-f)}\right)},$$

*where $\mathcal{R}_i^\lambda(\theta) := \mathbb{E}[\mathcal{L}_i^\lambda(\theta)], \forall \theta \in \Theta$, and $\beta := \sqrt{\mathrm{Pdim}(\mathcal{H}) \log\left(\frac{em}{\mathrm{Pdim}(\mathcal{H})}\right)} + \sqrt{\log(1/\delta)}$.*

*Proof.*

$$\mathcal{L}_i^\lambda(\theta) = (1 - \lambda)\mathcal{L}_i(\theta) + \lambda \mathcal{L}_\mathcal{C}(\theta)$$

$$= (1 - \lambda)\left(\frac{1}{m} \sum_{(x,y) \in S_i} \ell(h_\theta(x), y)\right) + \lambda \left(\frac{1}{|\mathcal{C}|} \sum_{j \in \mathcal{C}} \frac{1}{m} \sum_{(x,y) \in S_j} \ell(h_\theta(x), y)\right)$$

$$= \left(1 - \lambda + \frac{\lambda}{|\mathcal{C}|}\right)\left(\frac{1}{m} \sum_{(x,y) \in S_i} \ell(h_\theta(x), y)\right) + \lambda \left(\frac{1}{|\mathcal{C}|} \sum_{\substack{j \in \mathcal{C} \\ j \neq i}} \left(\frac{1}{m} \sum_{(x,y) \in S_j} \ell(h_\theta(x), y)\right)\right)$$

$$= \frac{1}{m|\mathcal{C}|}\left(\sum_{(x,y) \in S_i} |\mathcal{C}|\left(1 - \lambda + \frac{\lambda}{|\mathcal{C}|}\right)\ell(h_\theta(x), y) + \sum_{\substack{j \in \mathcal{C} \\ j \neq i}} \sum_{(x,y) \in S_j} \lambda\ell(h_\theta(x), y)\right). \tag{19}$$

Let us now consider the set of $m|\mathcal{C}|$ real-valued independent[6] random variables defined as $\left\{|\mathcal{C}|\left(1 - \lambda + \frac{\lambda}{|\mathcal{C}|}\right)\ell(h_\theta(x), y) \mid (x, y) \in S_i\right\} \cup \{\lambda\ell(h_\theta(x), y) \mid (x, y) \in S_j, \forall j \neq i\}$. As per Assumption 4, we know that $\ell$ takes its values in $[0, 1]$, hence this set is composed of $m$ random variables with values in $\left[0, |\mathcal{C}|\left(1 - \lambda + \frac{\lambda}{|\mathcal{C}|}\right)\right]$ and $(|\mathcal{C}| - 1)m$ others with values in $[0, \lambda]$. Finally, note that by definition $\mathbb{E}\left[\mathcal{L}_i^\lambda(\theta)\right] = \mathcal{R}_i^\lambda(\theta)$. Where the expectation is taken over the independent sampling of the local data sets $(S_i)_{i \in \mathcal{C}}$ from the local data distribution of the correct clients. Thus using Hoeffding inequality, we get

$$\mathbb{P}\left(|\mathcal{L}_i^\lambda(\theta) - \mathcal{R}_i^\lambda(\theta)| \geq \varepsilon\right) \leq 2 \exp\left(\frac{-2m|\mathcal{C}|^2\varepsilon^2}{\left(1 - \lambda + \frac{\lambda}{|\mathcal{C}|}\right)^2 |\mathcal{C}|^2 + (|\mathcal{C}| - 1)\lambda^2}\right). \tag{20}$$

---

[6]The data sets have been sampled independently of each other and are composed of points sampled i.i.d from the local data distributions.

By using the change of variables $\varepsilon' = \sqrt{\frac{|\mathcal{C}|^2}{\left(1-\lambda+\frac{\lambda}{|\mathcal{C}|}\right)^2 |\mathcal{C}|^2 + (|\mathcal{C}|-1)\lambda^2}} \varepsilon$, we can use the Pseudo-dimension union bound (e.g. Theorem 11.8 from (Mohri et al., 2018)) to get that with probability at least $1 - \delta$, $\forall \theta$

$$
\begin{aligned}
|\mathcal{L}_i^\lambda(\theta) - \mathcal{R}_i^\lambda(\theta)| &\leq 2\sqrt{\frac{\left(1-\lambda+\frac{\lambda}{|\mathcal{C}|}\right)^2}{m} + \frac{\lambda^2(|\mathcal{C}|-1)}{m|\mathcal{C}|^2}} \left(\sqrt{\mathrm{Pdim}(\mathcal{H})\log\left(\frac{em}{\mathrm{Pdim}(\mathcal{H})}\right)} + \sqrt{\log(1/\delta)}\right) \\
&\leq 2\sqrt{\frac{\left(1-\lambda+\frac{\lambda}{n-f}\right)^2}{m} + \frac{\lambda^2}{m(n-f)}} \left(\sqrt{\mathrm{Pdim}(\mathcal{H})\log\left(\frac{em}{\mathrm{Pdim}(\mathcal{H})}\right)} + \sqrt{\log(1/\delta)}\right).
\end{aligned}
$$

$\square$

### C.4  Proof of Theorem 1

**Theorem 1.** *Let assumptions 1, 2, 3, and 4 hold, and let $\Phi$ be a function such that (7) holds. Consider Algorithm 1 with learning rate $\eta = \frac{1}{2L}$, $\lambda \in [0,1]$, and assume the aggregation function $F$ to be $(f, \kappa)$-robust. Then, for any $\delta > 0, T \geq 1$, we have with probability at least $1 - \delta$ (over the choice of samples) that*

$$
\begin{aligned}
\mathcal{R}_i(\theta_i^T) - \mathcal{R}_i(\theta_i^*) &\leq \left(1 - \frac{\mu}{2L}\right)^T \frac{L}{\mu}\mathcal{L}_0 + \frac{5L\lambda^2\kappa G^2}{\mu^2} + 2\lambda\Phi(\mathcal{D}_i, \mathcal{D}_\mathcal{C}) \\
&\quad + 4\beta\sqrt{\frac{\left(1-\lambda+\frac{\lambda}{n-f}\right)^2}{m} + \frac{\lambda^2}{m(n-f)}},
\end{aligned}
\tag{8}
$$

*where $\beta := \sqrt{\mathrm{Pdim}(\mathcal{H})\log\left(\frac{em}{\mathrm{Pdim}(\mathcal{H})}\right)} + \sqrt{\log(1/\delta)}$ and $\mathcal{L}_{i,*}^\lambda := \min_{\theta \in \Theta} \mathcal{L}_i^\lambda(\theta)$.*

*Proof.* In order to control the quantity $\mathcal{R}_i(\theta_i^t) - \mathcal{R}_i(\theta_i^*)$, we can write:

$$
\begin{aligned}
\mathcal{R}_i(\theta_i^t) - \mathcal{R}_i(\theta_i^*) &\leq \mathcal{R}_i(\theta_i^t) - \mathcal{R}_i^\lambda(\theta_i^t) &\quad (1) \\
&+ \mathcal{R}_i^\lambda(\theta_i^t) - \mathcal{L}_i^\lambda(\theta_i^t) &\quad (2) \\
&+ \mathcal{L}_i^\lambda(\theta_i^t) - \mathcal{L}_i^\lambda(\widehat{\theta}_i^\lambda) &\quad (3) \\
&+ \mathcal{L}_i^\lambda(\widehat{\theta}_i^\lambda) - \mathcal{L}_i^\lambda(\theta_i^*) &\quad (4) \\
&+ \mathcal{L}_i^\lambda(\theta_i^*) - \mathcal{R}_i^\lambda(\theta_i^*) &\quad (5) \\
&+ \mathcal{R}_i^\lambda(\theta_i^*) - \mathcal{R}_i(\theta_i^*) &\quad (6),
\end{aligned}
\tag{21}
$$

where $\widehat{\theta}_i^\lambda$ is the minimizer in $\Theta$ of the empirical objective function $\mathcal{L}_i^\lambda$, which we assume is strongly convex.

**For lines** 1 **and** 6, we use the property of our loss function described in (7), to write

$$
(1) + (6) = \lambda\left(\mathcal{R}_i(\theta_i^t) - \mathcal{R}_\mathcal{C}(\theta_i^t)\right) + \lambda\left(\mathcal{R}_\mathcal{C}(\theta_i^*) - \mathcal{R}_i(\theta_i^*)\right) \leq 2\lambda\Phi(\mathcal{D}_i, \mathcal{D}_\mathcal{C}).
\tag{22}
$$

**For lines** 2 **and** 5, since Assumption (4) holds true, we can use the results from Lemma 2. Let $\beta := \mathrm{Pdim}(\mathcal{H})\log(2m|\mathcal{C}|) + \log(4/\delta)$. By applying Lemma 2 with $\delta/2$ for both (2) and (5), we get that with probability at least $1 - \delta$:

$$
(2) + (5) \leq 4\beta\sqrt{\left(\frac{\left(1-\lambda+\frac{\lambda}{|\mathcal{C}|}\right)^2}{m} + \frac{\lambda^2(|\mathcal{C}|-1)}{m|\mathcal{C}|^2}\right)}.
\tag{23}
$$

**For line** 3, recall that assumptions 1 and 2, that $\eta = \frac{1}{2L}$, and that $F$ is assumed to be $(f, \kappa)$-robust. Hence, Lemma 1 holds true. we use (16) in the proof of Lemma 1, to get

$$\mathcal{L}_i^\lambda\left(\theta_i^t\right) - \mathcal{L}_i^\lambda(\widehat{\theta}_i^\lambda) \leq \frac{5L\lambda^2\kappa G^2}{\mu^2} + \left(1 - \frac{\mu}{2L}\right)^t \frac{L}{\mu}\left(\mathcal{L}_i^\lambda(\theta_i^0) - \mathcal{L}_i^\lambda(\widehat{\theta}_i^\lambda)\right)$$

**For line** 4, we can simply disregard it as (4) contributes a negative amount to the inequality.

Finally, combining all the above inequalities we get

$$\mathcal{R}_i(\theta_i^t) - \mathcal{R}_i(\theta_i^*) \leq \frac{5L\lambda^2\kappa G^2}{\mu^2} + 2\lambda\Phi(\mathcal{D}_i, \mathcal{D}_\mathcal{C}) + 4\beta\sqrt{\left(\frac{\left(1 - \lambda + \frac{\lambda}{|\mathcal{C}|}\right)^2}{m} + \frac{\lambda^2(|\mathcal{C}| - 1)}{m|\mathcal{C}|^2}\right)} \quad (24)$$

$$+ \left(1 - \frac{\mu}{2L}\right)^t \frac{L}{\mu}\left(\mathcal{L}_i^\lambda(\theta_i^0) - \mathcal{L}_i^\lambda(\widehat{\theta}_i^\lambda)\right).$$

We conclude by substituting $|\mathcal{C}| = n - f$.

$\square$

### C.5  Proof of (9)

**Equation** (9)

$$\lambda^* \approx \Pi_{[0,1]}\left(\frac{\sqrt{\frac{\mathrm{Pdim}(\mathcal{H})}{m}} - \Phi(\mathcal{D}_i, \mathcal{D}_\mathcal{C})}{\frac{f}{n}G^2}\right),$$

*Proof.* Here, our goal is to reasonably approximate the value of lambda $\lambda^*$ for which the right-hand side of (24) is minimized. To do so, we first ignore the asymptotically negligible terms. Essentially, we consider that $T$ is large enough so that the term $\left(1 - \frac{\mu}{2L}\right)^T \frac{L}{\mu}\left(\mathcal{L}_i^\lambda(\theta_i^0) - \mathcal{L}_i^\lambda(\widehat{\theta}_i^\lambda)\right)$ is close enough to zero. In the reminder, we denote by $\theta_i^\infty$ the output of Algorithm 1 for $T \to \infty$. Even after ignoring this term, seeking the optimal value $\lambda^*$ is the solution to a 4th-degree equation, which may not admit a close-form solution in general. To make the computation of an approximate minimizer more tractable, we proceed with the following simplification using the fact that $\sqrt{a + b} \leq \sqrt{a} + \sqrt{b}$ for any $a, b \geq 0$:

$$\mathcal{R}_i(\theta_i^\infty) - \mathcal{R}_i(\theta_i^*) \leq \alpha\lambda^2 + 2\lambda\Phi(\mathcal{D}_i, \mathcal{D}_\mathcal{C}) + 4\frac{1 - (1 - \frac{1}{|\mathcal{C}|})\lambda}{\sqrt{m}}\beta + 4\frac{\lambda\sqrt{|\mathcal{C}| - 1}}{|\mathcal{C}|\sqrt{m}}\beta, \quad (25)$$

with $\alpha := \frac{6L\kappa G^2}{\mu^2}$. Now the right-hand side is a quadratic function of $\lambda$ which can be easily minimized. Specifically, the solution of (25) is

$$\lambda^* = 2\left(1 - \frac{1}{|\mathcal{C}|}\right)\frac{\beta}{\alpha\sqrt{m}} - 2\frac{\sqrt{|\mathcal{C}| - 1}}{|\mathcal{C}|\alpha\sqrt{m}}\beta - \frac{\Phi(\mathcal{D}_i, \mathcal{D}_\mathcal{C})}{\alpha} \quad (26)$$

Assuming $|\mathcal{C}|$ is large enough, we simplify the above as

$$\lambda^* \approx \frac{2\beta}{\alpha\sqrt{m}} - \frac{\Phi(\mathcal{D}_i, \mathcal{D}_\mathcal{C})}{\alpha}$$

$$\approx \frac{1}{\alpha}\left(\frac{2\beta}{\sqrt{m}} - \Phi(\mathcal{D}_i, \mathcal{D}_\mathcal{C})\right) \quad (27)$$

$\square$

# D   Experimental Details

In this section, we give the implementation details of our experiments for each setting and we provide some additional figures. For all the experiments, we run Algorithm 1 with full gradients for a number $T$ of iterations which we change depending on the dataset.

## D.1   Compute

We run our experiments on a server with the following specifications:

- HPe DL380 Gen10
- 2 x Intel(R) Xeon(R) Platinum 8358P CPU @ 2.60GHz
- 128 GB of RAM
- 740GB ssd disk
- 2 Nvidia A10 GPU cards

For most of the experiments, we only use one of the two GPUs.

Using these compute resources, each experiment on the MNIST dataset (meaning each subfigure in Figure 2(bottom)) took less than 72 hours to run. Each experiment on Phishing dataset took less than 8 hours to run.

## D.2   MNIST

For the experiments on MNIST, we use the following Convolutional Neural Network:

- Convolutional Layer (1, 32, 5, 1) + ReLU + Maxpooling
- Convolutional Layer (32, 64, 5, 1) + ReLU + Maxpooling
- Fully Connected Layer (4096, 1024) + ReLU
- Fully Connected Layer (1024, 10)

We use $T = 100$ and the learning rate $\eta = 0.05$.

## D.3   Binary MNIST

We run some experiments on a binarized version of the MNSIT dataset, where the task is to output whether or not the MNIST class of the image, which corresponds to a digit between $0$ and $9$, is strictly smaller than $5$.

We use $T = 150$ and the learning rate $\eta = 0.002$.

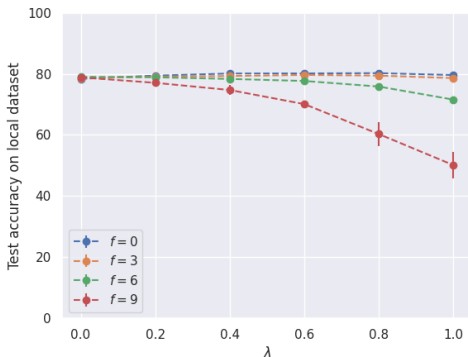

Figure 3: Effect of Byzantine fraction. Binary MNIST with logistic regression. $m = 256, \alpha = 3$

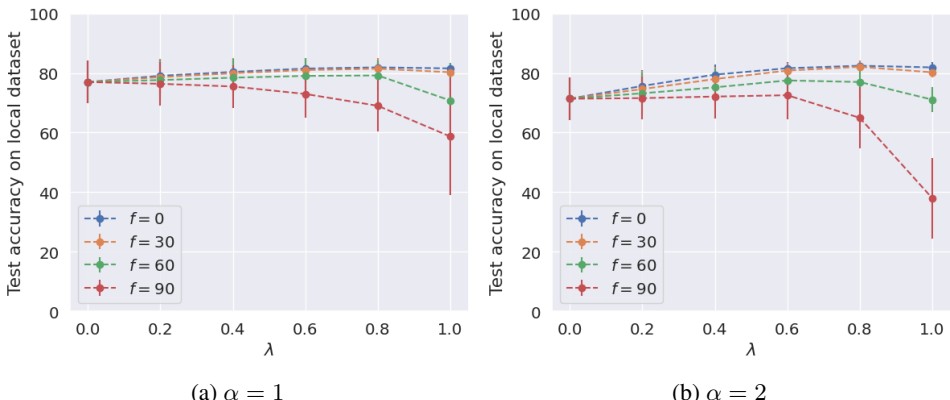

(a) $\alpha = 1$            (b) $\alpha = 2$

Figure 4: Effect of adversarial fraction and heterogeneity. Binary MNSIT dataset with logistic regression. $n = 200, m = 32$.

### D.4 Phishing

We use $T = 500$ and the learning rate $\eta = 0.1$.

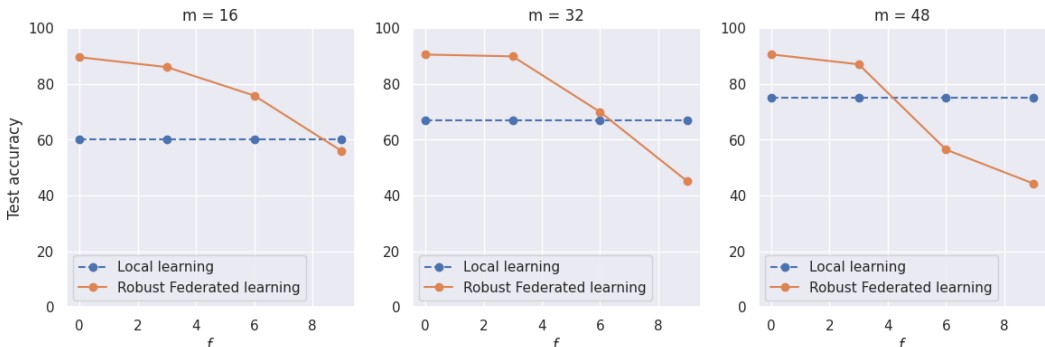

Figure 5: Local Vs FL performance on local test dataset. Phishing dataset with $n = 20, \alpha = 3$. As the number of local samples increases, the Byzantine fraction threshold above which local learning performs better than Robust Federated Learning gets smaller.

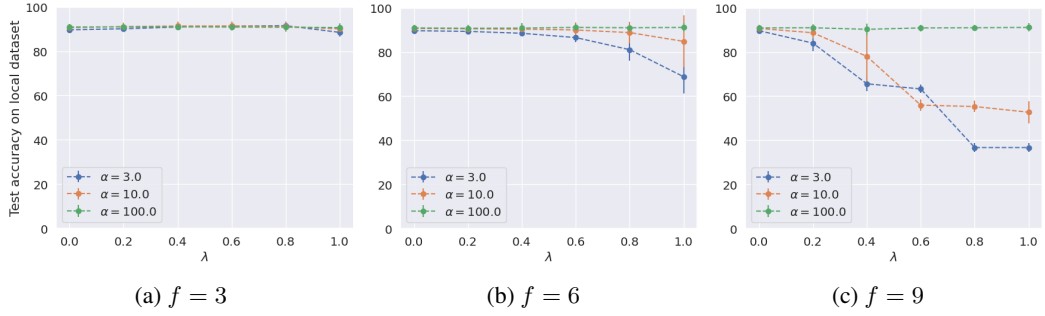

(a) $f = 3$         (b) $f = 6$         (c) $f = 9$

Figure 6: Effect of adversarial fraction and heterogeneity. Phishing dataset with logistic regression. $n = 20, m = 128$. $\alpha = 100$ corresponds practically to the homogeneous case.

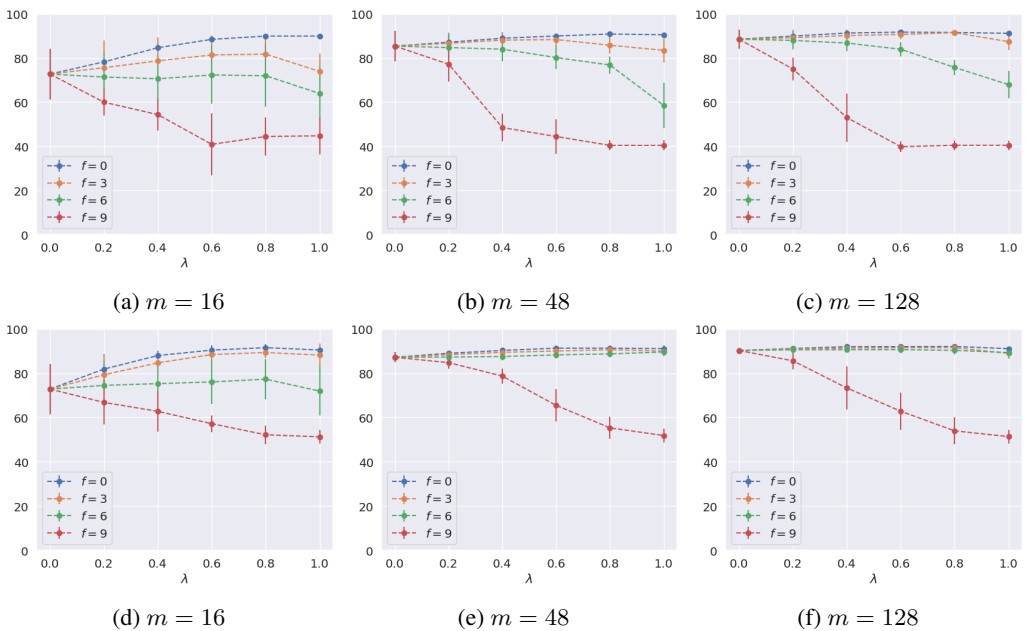

Figure 7: Effect of the adversarial fraction and the data size. Phishing with logistic regression. $n = 20$, Auto-FOE attack, (top) $\alpha = 3$, (bottom) $\alpha = 10$.

