# OpenReview forum: "Fine-Tuning Personalization in Federated Learning to Mitigate Adversarial Clients"
_NeurIPS.cc/2024/Conference — NeurIPS 2024 poster_

### Official Review · Reviewer_Kq44 · 2024-06-22

**Soundness:** 3
**Presentation:** 2
**Contribution:** 3
**Rating:** 5
**Confidence:** 3

**Summary:**

This paper proposes a fundamental validation to understand the relationship between local models and the global model to mitigate the impact of adversarial clients. The level of collaboration needs to be chosen carefully because of the existence of adversarial clients. The theoretical analysis is provided to validate the statement, considering data heterogeneity, the fraction of adversarial clients, and data scarcity. Several simulated and open-source datasets are used to further demonstrate the effectiveness of the method.

**Strengths:**

This paper proposes a simple yet easily applicable method to mitigate Byzantine adversaries in personalized FL, supported by thorough theoretical proof.

**Weaknesses:**

1. Some typos, such as the missing space behind "Section 2" in Line 112.
2. Although the theoretical proof is thorough, more experiments on different datasets, Byzantine attack methods, and defense methods should be evaluated.
3. For the simulated datasets in Section 2.2, cross-device (n=600/f=100) is employed, but the experimental validation in Section 3.3 uses cross-silo (n=20/f=0,3,6,9).
4. The models used for each dataset are not mentioned.
In summary, although theoretical proof is provided, the practical applicability of the method has not been sufficiently demonstrated.

**Questions:**

Refer to Weaknesses.

**Limitations:**

yes

---

> ### Author Rebuttal · Authors · 2024-08-07
>
> We thank the reviewer for the positive feedback and address the reviewer's comments point by point.
>
> 1- We thank the reviewer for pointing out the typos.
>
> 2- We did use different attacks in our experiments but did not notice any important differences. For the defense, we only considered the most state-of-the-art one (NNM pre-aggregation rule [1] followed by trimmed mean). We will clarify this in the final version of the paper.
>
> 3- The reviewer is correct in remarking that our numerical validation experiments cover both cross-device and cross-silo scenarios. We did run a few experiments in the cross-device setting with binary MNIST (Figure 4 in the appendix for instance), and we obtained remarkably similar conclusions (which is intuitive as the bounds dependence on $n$ is not explicit). We merely reported the cross-silo results in the main text as they were more extensive, spanning many heterogeneity levels and data sample sizes.
>
> 4- We thank the reviewer for the remark on the models used. In fact, we have provided the models used for each dataset in Appendix D but we understand that this might not be clear from the main text. We will add a reference to this part of the appendix in the experiments figure caption to make that clear.

---

> > ### Comment · Reviewer_Kq44 · 2024-08-08
> > **Rating**
> >
> > Given the authors’ response, I will maintain my rating.

---

### Official Review · Reviewer_cJWD · 2024-07-10

**Soundness:** 3
**Presentation:** 3
**Contribution:** 2
**Rating:** 5
**Confidence:** 3

**Summary:**

This paper studies fine-tuning personalization in federated learning (FL) to mitigate the impact of adversarial clients. The authors leverage interpolation techniques for personalization, and they derive the closed-form approximation of the interpolation parameter $\lambda$. The study comprehensively considers both data heterogeneity and the presence of adversarial clients in the context of tailoring personalized FL.

**Strengths:**

1. The authors consider that fine-tuning personalization in FL can mitigate the impact of adversarial clients, which extends existing Byzantine adversaries in FL.
2. They derive the closed -form approximation of the interpolation parameter $\lambda$, which can guide the fine-tuning procedure.
3. The theoretical analysis is comprehensive.

**Weaknesses:**

1. The proposed fine-tuning personalization strategy requires each client should broadcast its model. Besides, each client should send the gradients to other clients. This is not efficient and may incur other privacy issues.
2. The prediction tasks in this paper are simple.
Other issues:
1. Then then --> Then the in 142.

**Questions:**

1. For more convincing, the authors should consider other more complicated datasets.
2. The suggested fine-tuning strategy for personalization necessitates that each client share its gradients with others, which could potentially raise privacy concerns. Moreover, clients might be able to identify adversarial clients through the gradients accumulated during communication.

**Limitations:**

N/A.

---

> ### Author Rebuttal · Authors · 2024-08-07
>
> We thank the reviewer for the positive feedback and address the reviewer's questions below.
>
> **On the datasets.**
> Our contribution is mainly theoretical, and we provide some experimental results (covering mean estimation, binary, and multi-class classification) to convey the meaningfulness of our bounds. We understand that a more thorough empirical study can be of benefit to the community, but we believe this is out of the scope of this paper and we leave it for future work.
>
> **On non-efficiency and privacy.**
> We thank the reviewer for this essential point. Privacy is of key importance and, indeed, the proposed algorithm does not address this issue. In this work, we focus on the trade-off between generalization and robustness, which we see as fundamental and essential to answering the question of whether or not to collaborate in the first place.
>
> However, recent literature has suggested that personalization can help improve privacy-utility trade-offs in some cases~[1,2]. The frameworks used in these works are different from ours (split-model personalization for the former and multitask learning for the latter), but we believe that an interesting future direction would be to link (one of) these frameworks to ours in order to characterize simultaneously the trilemma of privacy, utility, and robustness.
>
> We will add the privacy limitation in the paper for more clarity.
>
>
> [1] Bietti, A., Wei, C.-Y., Dudík, M., Langford, J., and Wu, Z. S. (2022). Personalization improves privacy-accuracy tradeoffs in federated learning.
>
> [2] Liu, Ziyu and Hu, Shengyuan and Wu, Zhiwei Steven and Smith, Virginia (2022). On privacy and personalization in cross-silo federated learning. Proceedings of the 36th International Conference on Neural Information Processing Systems.

---

> > ### Author Response · Authors · 2024-08-12
> >
> > We hope our response has addressed your doubts and concerns. In which case, we kindly urge you to reconsider the rating of our paper accordingly. We remain at your disposal for clarifying any additional concerns.
> >
> > We are thankful for your time and effort in reviewing our paper!

---

### Official Review · Reviewer_Fsfa · 2024-07-28

**Soundness:** 2
**Presentation:** 3
**Contribution:** 2
**Rating:** 5
**Confidence:** 3

**Summary:**

This paper considers an FL setting where some clients can be adversarial, and we derive conditions under which full collaboration fails. Specifically, they analyze the generalization performance of an interpolated personalized FL framework in the presence of adversarial clients. The authors claim that they precisely characterize situations when full collaboration performs strictly worse than fine-tuned personalization.

**Strengths:**

The idea is intuitive and easy to understand. With the presence of adversarial, less collaboration should work better. In addition, this paper proposed a new formulation for personalized FL, combining local loss and global loss.

**Weaknesses:**

Section 2 doesn't make sense to me. Proposition 2 characterizes the difference between two variables, a local variable $\mu_i$, and a variable depending on collaboration (y^{\lambda}).
This manuscript only considers deterministic cases. The assumption is strong, such as assumptions 2 and 4.
The bound shown in the analysis is loose and the conclusion does not convince me.

**Questions:**

In equation 9, when there is less data heterogeneity, \Psi() ->0 and G ->0, $lambda$  -> 1. Why do we need to collaborate when the data is homogeneous? I think we can train locally to avoid the adversary.

Is assumption 3 necessary? Is it repetitive with Assumption 1?

Is the model only effective for binary classification problems?

**Limitations:**

Experiments are a bit simple.  Assumptions are strong.

---

> ### Author Rebuttal · Authors · 2024-08-07
>
> We thank the reviewer for the feedback and address the reviewer's questions below.
>
> **On Section 2.**
> We are not certain what exactly bothers the reviewer in Section 2. Perhaps the relation to the general problem was unclear, in which case the following paragraph might help make it clearer. We will also clarify this further in the paper.
>
> In Section 2, we consider the special learning problem of personalized mean estimation, which is an instantiation of the general personalized learning problem (3) introduced in the paper. In this problem, for each client $i$, for a sample $y_i$ drawn from a distribution $\mathcal{D}_i$, the point-wise loss function $\ell(\theta, y_i)$ is given by $\lVert y_i - \theta \rVert^2$.
>
> The local risk function $R_i(\theta):= \mathbb{E}_{y_i  \sim \mathcal{D}_i} \lVert y_i - \theta \rVert^2$ is minimized at $\theta^*_i = \mu_i$, where $\mu_i$ is the mean of the distribution $\mathcal{D}_i$.
> Consequently, for any $\theta$, $R_i(\theta) - R_i(\theta^*_i) = \lVert \theta - \mu_i \rVert^2$ is the distance to the true local distribution mean. One possible candidate for $\theta$ is the empirical mean of client $i$'s data points (local estimator). Another possibility is the empirical mean of all the aggregated clients' data points (global estimator). In our paper, we analyze a third option which is the $\lambda$-interpolated estimator.
>
> Proposition 1 presents a bound on the error of the latter, i.e., $R_i(\theta) - R_i(\theta^*_i) = \lVert \theta - \mu_i \rVert^2$, for $\theta = y^{\lambda}_i$, $y^{\lambda}_i$ being the solution of the $\lambda$-interpolated empirical loss minimization problem in this case, defined in (4).
>
> **On the strength of assumptions 2 and 4**
> Assumption 2 is standard in the Byzantine and the Federated Learning literature (e.g., see[1, 4]). Additionally, it is necessary since the absence of this assumption (i.e. G=+$\infty$) leads to an unbounded error as shown in [1]. Assumption 4 is standard in the learning theory literature (e.g., see [2,3]) and the assumption could be re-defined considering boundedness by some positive parameter instead of $1$ without loss of generality. It could also be replaced by assuming that the loss is Lipschitz since the parameter space is bounded, which is also standard in learning theory.
>
> **Why do we need to collaborate when the data is homogeneous?**
> This is a very important question indeed. The intuition suggested by our results is twofold. First, if the participants do not have enough data locally, collaborating might be beneficial even in the presence of Byzantine attackers, since otherwise, the model trained on local data can be of poor quality. Second, if the heterogeneity is small enough, the effect of Byzantine players on the training is smaller, since theoretically the accuracy loss due to adversaries is linked to the term $\frac{f}{n} G^2$.
>
> **Is Assumption 3 necessary? Is it repetitive with Assumption 1?**
> Assumption 3 is not repetitive with Assumption 1. While Assumption 1 ensures Lipschitz smoothness and strong convexity of the loss functions, Assumption 3 ensures that the interior of the parameter search space $\Theta$ contains a minimizer of the loss functions. These are two separate conditions on the learning problem.
>
> **Is the model only effective for binary classification problems? Can the analysis be extended to the stochastic case?**
>  The analysis can be generalized beyond binary classification to all settings covered by the VC dimension or the pseudo-dimension generalization theory,  including regression and multi-class classification. Moreover, the learning guarantees we provide are not deterministic but they already account for the random choice of the training samples, e.g., see Theorem 1. By the stochastic case, does the reviewer mean the stochastic gradient descent-based methods?
>
>
> [1] Allouah, Y., Farhadkhani, S., Guerraoui, R., Gupta, N., Pinot, R., and Stephan, J. (2023). Fixing by mixing: A recipe for optimal byzantine ml under heterogeneity. In International Conference on
> Artificial Intelligence and Statistics, pages 1232–1300. PMLR
>
> [2] Blitzer, J., Crammer, K., Kulesza, A., Pereira, F., and Wortman, J. (2007). Learning bounds for domain adaptation. In Platt, J., Koller, D., Singer, Y., and Roweis, S., editors, Advances in Neural Information Processing Systems, volume 20. Curran Associates, Inc.
>
> [3] Mohri, M., Rostamizadeh, A., and Talwalkar, A. (2018). Foundations of machine learning.
>
> [4] Sai Praneeth Karimireddy, Satyen Kale, Mehryar Mohri, Sashank J. Reddi, Sebastian U. Stich, Ananda Theertha Suresh. SCAFFOLD: Stochastic Controlled Averaging for Federated Learning

---

> ### Comment · Reviewer_Fsfa · 2024-08-09
>
> Thanks for your reply. Based on this response, I checked the manuscript again. I have raised the score from 4 to 5.

---

### Official Review · Reviewer_uEJH · 2024-07-31

**Soundness:** 3
**Presentation:** 3
**Contribution:** 3
**Rating:** 7
**Confidence:** 3

**Summary:**

This paper presents theoretical analysis and experimental validation results of the allowed level of collaboration in personalized FL with the presence of a fraction of Byzantine adversaries.

**Strengths:**

+ This paper targets a very important and challenging problem in the personalized FL settings.
+ The theoretical analysis and results are analytically rigorous and thorough.
+ The experimental validation results are also comprehensive and well complement with the theoretical analysis.
+ The results correlating the allowed level of collaboration and the tolerable fraction of adversaries are particularly appreciated.

**Weaknesses:**

- The experimental validations can still be further improved from multiple aspects. For example, it may not be very convincing by using simulated datasets being generated by simple 1D sampling. The data heterogeneity settings should be more complicated and practical accordingly. More complicated models should also be used.
- The analysis results only apply to the simple problem of binary classification. What's its generalizability to more practical multi-class classification?

**Questions:**

See weakness above.

**Limitations:**

See weakness above.

---

> ### Author Rebuttal · Authors · 2024-08-07
>
> We thank the reviewer for the positive feedback and constructive comments. We address below the reviewer's comments.
>
> **On the experiments.**
> Since our contribution is mainly theoretical, we provide some experimental results mainly to convey the meaningfulness of our bounds. In Section 2, we restricted our experiments to the 1-dimensional case both for simplicity and scalability reasons. Indeed, with a 1-dimensional dataset, it was possible to consider a large number of clients ($600$) and run these experiments multiple times to obtain meaningful confidence intervals. In Section 3, we used Dirichlet sampling to simulate several heterogeneity levels in the classification case.  We chose this technique, as it is a common method for testing an algorithm with a controlled level of heterogeneity in the Byzantine Learning literature. In fact, we mainly used this technique as it allows to navigate between two important scenarios: i) homogeneity when $\alpha \rightarrow \infty$ and ii) extreme heterogeneity when $\alpha \rightarrow 0$.  We agree that in future exploration of our scheme, other heterogeneous data-generating techniques can be used.
>
> **On the generalizability to more practical multi-class classification.**
> Our analysis can be generalized beyond binary classification to all settings covered by the VC dimension or the pseudo-dimension generalization theory, including for instance regression and multi-class classification. We will clarify this in the paper.

---

> > ### Comment · Reviewer_uEJH · 2024-08-12
> >
> > Thanks for the rebuttal. I will keep my score.

---

### Decision · Program_Chairs · 2024-09-25

**Decision:**

Accept (poster)

**Comment:**

This paper provides a nice analysis and discussion on the interplay between collaboration, personalization, and adversarial clients in federated learning. The focus is on the theory, with an intuitive special case of mean estimation presented in Section 2 and the full analysis of the generalization bound presented in Section 3. The theoretical findings have been complemented by experiments which show useful insights. The conclusions and avenues of future work have been well-discussed too. As a theory-focused paper, it would be nice if the technical challenges and novelty in the methodology can be explained more.